# PhysWorld: From Real Videos to World Models of Deformable Objects via Physics-Aware Demonstration Synthesis

## Abstract

Interactive world models that simulate object dynamics are crucial for robotics, VR, and AR. However, it remains a significant challenge to learn physics-consistent dynamics models from limited real-world video data, especially for deformable objects with spatially-varying physical properties. To overcome the challenge of data scarcity, we propose PhysWorld, a novel framework that utilizes a simulator to synthesize physically plausible and diverse demonstrations to learn efficient world models. Specifically, we first construct a physics-consistent digital twin within MPM simulator via constitutive model selection and global-to-local optimization of physical properties. Subsequently, we apply part-aware perturbations to the physical properties and generate various motion patterns for the digital twin, synthesizing extensive and diverse demonstrations. Finally, using these demonstrations, we train a lightweight GNN-based world model that is embedded with physical properties. The real video can be used to further refine the physical properties. PhysWorld achieves accurate and fast future predictions for various deformable objects, and also generalizes well to novel interactions. Experiments show that PhysWorld has competitive performance while enabling inference speeds 47 times faster than the recent state-of-the-art method, *i.e.*, Phys-Twin. The code and pre-trained models will be publicly available.

## 1 Introduction

Humans possess the ability to predict object movements to achieve specific goals during interaction, which is developed through accumulated experience from observations and experiments. This skill is not only fundamental to human intelligence but also a crucial requirement for various technological applications, including robotics, virtual reality (VR), and augmented reality (AR). Thus, it becomes increasingly popular to construct world models for dynamics modeling of objects.

Existing works explore this mainly from learning-based and physics-based simulation perspectives. **First**, learning-based methods generally take neural networks such as Graph Neural Networks (GNN) (Sanchez-Gonzalez et al., 2020) and Multilayer Perceptron (MLP) (Zhu et al., 2024) for dynamics modeling. Such models can achieve real-time inference and be applicable to a variety of objects, including plasticine, cloth, and fluids. However, they rely on extensive training data represented by particle (Wang et al., 2023), mesh (Pfaff et al., 2020), and 3D point (Zhang et al., 2025b). Although the data can be derived from either simulators or real videos, the simulated data can be physically inconsistent with real-world ones, and sufficient real-world data are labor-intensive to acquire. Recently, AdaptiGraph (Zhang et al., 2024a) further introduces the physical property-conditioned GNN to adapt unseen objects during interaction, but its physics-inconsistent data synthesis and global physical parameters restrict applicability to objects with spatially varying properties. **Second**, physics-based simulation methods (Zhang et al., 2024c; Huang et al., 2024a; Zhong et al., 2024; Lin et al.) model object dynamics using established simulators such as the Material Point Method (MPM) (Jiang et al., 2016; Stomakhin et al., 2013; Bardenhagen et al., 2000) or the Mass-Spring System (MSS), often in conjunction with the optimization of physical properties of objects. For instance, recent PhysTwin (Jiang et al., 2025) leverages the way to construct digital twins of objects from sparse videos, enabling effective resimulation. Benefiting from the physics-

based prior knowledge embedded in the simulator, these methods can achieve relatively realistic simulations. Nevertheless, their real-time inference capability is still limited.

It can be seen that large amounts of data or powerful simulators are required for the accuracy of world models, while lightweight modeling manners (*e.g.*, GNN) are necessary for the efficiency. When real-world observation sequences are short, it remains challenging to construct both accurate and fast world models for deformable objects. To address this issue, we propose to build a bridge between powerful simulators and lightweight modeling manner, utilizing simulators to synthesize a large amount of data to learn a lightweight world model. Therein, we argue that the *physical plausibility* and *diversity* of synthesized data are crucial to obtain satisfactory performance, thus propose a physics-consistent digital twin construction and diverse demonstration generation methods.

Specifically, we propose a framework named PhysWorld. It consists of three main stages. **Firstly**, to establish a physics-consistent digital twin, we first utilize a Vision-Language Model (VLM) to automatically select the optimal constitutive models for deformable objects within the MPM simulation. We then introduce a global-to-local optimization strategy to refine physical properties (*e.g.*, friction, density, and Young's modulus), ensuring the MPM simulations align with the observed video. **Secondly**, since the motion trajectory of the real video is single and the learned physical parameters of the digital twin inevitably exist errors, we propose Various Motion Pattern Generation (VMP-Gen) and Part-aware Physical Property Perturbation ($P^3$-Pert) methods. It enables synthesizing extensive and diverse 4D demonstrations. **Thirdly**, the demonstrations are used to train a GNN-based world model embedded with spatially varying physical properties. The original real videos can also be used to fine-tune the physical property values, thereby enhancing the alignment between GNN models and real-world object dynamics.

The learned world model can perform accurate and fast future prediction, and also generalizes well to novel interactions. Experiments on 22 scenarios show our model has competitive performance while enabling more efficient inference. In particular, it is 47 times faster than the recent state-of-the-art method, *i.e.*, PhysTwin (Jiang et al., 2025).

Our contributions can be summarized as follows:

(1) We propose a framework that constructs not only accurate but also fast world models for deformable objects from short real-world videos via physics-consistent digital twin construction and diverse demonstration generation, named PhysWorld.

(2) We propose the Various Motion Pattern Generation (VMP-Gen) and Part-aware Physical Property Perturbation ($P^3$-Pert) methods for the diversity of synthesized demonstrations.

(3) Experiments on 22 scenarios show our model has competitive performance while enabling inference speeds 47 times faster than the recent state-of-the-art method, *i.e.*, PhysTwin.

## 2 RELATED WORK

### 2.1 PHYSICS-BASED SIMULATION OF DEFORMABLE OBJECTS

Recent advances in dynamic scene reconstruction (Li et al., 2022; Wu et al., 2024a; Luiten et al., 2024; Wu et al., 2024b) have integrated physics-based simulators with modern 3D representations such as NeRF (Mildenhall et al., 2021) and 3DGS (Kerbl et al., 2023), utilizing their physical mechanics modeling to learn dynamics grounded in physical principles. For example, PIE-NeRF (Feng et al., 2024) combines nonlinear elastodynamics with NeRF to generate plausible animations. PhysGaussian (Xie et al., 2024) integrates material point methods (MPM) (Jiang et al., 2016; Stomakhin et al., 2013; Bardenhagen et al., 2000) with 3DGS for interactive object representations. VR-GS (Jiang et al., 2024) employs extended position-based dynamics alongside 3DGS to develop physics-aware virtual reality systems. However, these methods require manually specified simulation parameters, risking a mismatch with real-world observations. Subsequent research (Chen et al., 2022a; Li et al., 2023; Qiao et al., 2022; Zhang et al., 2024c; Zhong et al., 2024; Huang et al., 2024a) addresses this via system identification, estimating parameters directly from video during reconstruction. PhysTwin (Jiang et al., 2025) exemplifies this, using a mass-spring model optimized from interaction videos to enable action-conditioned prediction of elastic objects.

While physics-grounded simulations offer greater realism through inherent physical priors, they face challenges. High-fidelity simulators like MPM incur prohibitive computational costs, hindering real-time applications. A persistent domain gap also exists between the dynamics of simplified simulators and complex real phenomena, frequently causing motion prediction inaccuracies.

## 2.2 Learning-Based Simulation of Deformable Objects

Learning dynamics models directly from data has emerged as a prominent research direction (Sanchez-Gonzalez et al., 2020; Chen et al., 2022b; Evans et al., 2022; Ma et al., 2023; Wu et al., 2019; Xu et al., 2019; Zhang et al., 2024a; 2025a; 2024b). Compared to traditional physics-based approaches, these methods can utilize lightweight neural networks to achieve significantly greater efficiency. In particular, Graph-based neural networks (GNNs) have proven particularly effective at learning dynamics across diverse deformable objects, including plasticine (Shi et al., 2023), cloth (Lin et al., 2022; Pfaff et al., 2020), and fluids (Li et al., 2018; Sanchez-Gonzalez et al., 2020). For instance, GS-Dynamics (Zhang et al., 2024b) employs tracking and appearance priors derived from Dynamic Gaussians (Luiten et al., 2024) to train GNN-based world models on real-world interaction videos with deformable objects.

However, learning-based methods require vast data and lack generality. In this work, our PhysWorld synergizes physics and learning-based methods. PhysWorld uses the MPM simulator as a data factory, generating physically plausible and diverse demonstrations from real videos to train a GNN-based world model. It can take a better trade-off between accuracy and efficiency.

## 3 PhysWorld

PhysWorld builds a physics-consistent MPM-based digital twin from short real-world videos by leveraging a VLM to determine the appropriate constitutive models and a global-to-local physical property optimization strategy to align the simulation with real observations. The calibrated twin is then used to generate a variety of interaction data, which trains a GNN-based world model enabling real-time simulation. The physical properties embedded in GNNs are finally fine-tuned on the original real data to mitigate the sim-to-real gap.

### 3.1 Physics-Consistent Digital Twin

Empirical research (Zhang et al., 2024b; Cao et al., 2024; Cai et al., 2024) reveals that short real interaction videos alone are insufficient for learning comprehensive world models of deformable objects, whereas estimating physical properties proves comparatively tractable. This motivates our proposition, *i.e.*, leveraging a high-fidelity physics engine to impose fundamental physical constraints (*e.g.*, mass and momentum conservation laws) as strong priors. Rather than learning physics from first principles, we parameterize the object within the physics engine and optimize only its digital twin's constitutive model and physical properties using the real data.

**Construction of Initial Digital Twin** Following object point cloud extraction from real interaction videos (Jiang et al., 2025), we construct a physics-consistent MPM-based digital twin. To ensure simulation fidelity under continuum mechanics principles, we utilize a tetrahedral mesh refinement method (Wang et al., 1996) to fill interior voids in the extracted point cloud, thereby creating complete volumetric representations. We implement two robotic manipulation primitives within the MPM framework: (1) **Grasping**, which is achieved by directly assigning velocities to the Eulerian grids near the contact points; and (2) **Pushing**, which is implemented via geometric controllers defined by a Signed Distance Function (SDF) that transfer motion to the proximal grid nodes. Meanwhile, the videos are analyzed using Qwen3 (Yang et al., 2025) to automatically identify the most suitable material constitutive models from our physics library. This VLM-based method dynamically adapts material representations according to the observed deformation behaviors. The constitutive models implemented and the associated prompts are provided in the Appendix B.1.

**Physical Property Optimization** To ensure physics-consistent alignment between our digital twin and real-world objects, we optimize the twin's physical properties (*e.g.*, friction coefficients, density, and Young's modulus) using real-world observational data. More specifically, the densified point cloud is registered as MPM particles. Using the implemented manipulation primitives, we

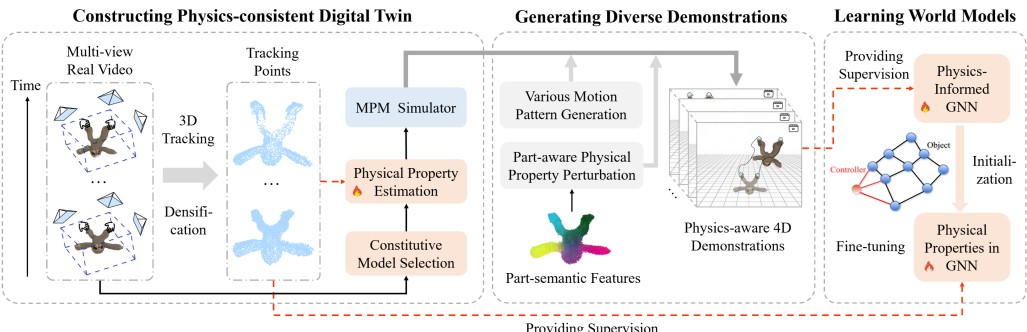

Figure 1: Overview of PhysWorld. The framework first constructs a physics-consistent digital twin from videos, then uses it to generate diverse 4D demonstrations, and finally trains a GNN-based world model for real-time future state prediction.

drive the system to re-simulate the hand motion trajectory extracted from real-world data. The entire MPM simulation framework is differentiable, enabling the computation of gradients of the Chamfer Distance and $\mathcal{L}_1$ loss between the simulated outcomes and the ground truth. This differentiability facilitates the optimization of the physical properties of MPM particles to achieve enhanced alignment. The loss function used for optimization is provided in Appendix B.2. Experimental results show that parameter initialization critically affects optimization performance: well-chosen initial values lead to faster convergence and higher performance. Thus, we employ a global-to-local optimization strategy. First, on the global stage, we optimize homogeneous physical properties across all particles. Then, on the local stage, we refine per-particle heterogeneous properties.

## 3.2 Augmented Interaction Demonstration Synthesis

To train a robust GNN-based world model, we generate various interaction demonstrations by simultaneously varying motion patterns and applying part-aware perturbations to physical properties.

**Various Motion Pattern Generation** Addressing the need for complex motion patterns during inference, control point trajectories are generated via curvature-constrained Bézier (Prautzsch et al., 2002) curves with integrated velocity regimes. One spatial trajectory $\boldsymbol{x}(t)$ can be defined by a cubic Bézier curve, *i.e.*,

$$\boldsymbol{x}(t) = \boldsymbol{B}(u(t)), \tag{1}$$
$$\boldsymbol{B}(u) = (1-u)^3\boldsymbol{p}_0 + 3(1-u)^2 u\boldsymbol{p}_1 + 3(1-u)u^2\boldsymbol{p}_2 + u^3\boldsymbol{p}_3. \tag{2}$$

$\boldsymbol{p}_0$ and $\boldsymbol{p}_3$ are start and end points of the trajectory. $\boldsymbol{p}_1$ and $\boldsymbol{p}_2$ are curvature control points generated through randomized curvature parameters. $u \in [0, 1]$ is the Bézier parameter. The temporal evolution of the trajectory is governed by the time-to-parameter mapping $u(t)$, which is constructed using normalized arc-length parameterization (Hartlen & Cronin, 2022). The normalized arc length $s(t)$ can be formulated as:

$$s(t) = \frac{\int_0^t v(\tau)d\tau}{\int_0^T v(\tau)d\tau}, \tag{3}$$

where $v(t)$ represents the velocity profile along the trajectory. This ensures smooth and physically consistent motion along the curve. To enable diverse and natural motion generation, we implement a three-phase velocity profile consisting of acceleration, uniform motion, and deceleration phases. Crucially, these phases are connected with smooth $C^1$ transitions, ensuring continuous velocity and avoiding abrupt changes in acceleration.

**Part-aware Physical Property Perturbation** To address potential inaccuracies in physical properties optimized in MPM solely from short real videos, we impose semantic-partition-guided stochastic perturbations on the optimized physical properties during demonstration synthesis. This strategy incorporates controlled diversity in physical property distribution while preserving continuity to match real-world distributions and maintain simulation stability. We extract the part-semantic feature vector $\{\boldsymbol{F}_i\}_{i=1}^N$ for each of the $N$ MPM particles via PartField (Liu et al., 2025). The feature similarity between particles $i$ and $j$ is computed as,

$$S_{ij} = \exp\left(-\frac{\|\boldsymbol{F}_i - \boldsymbol{F}_j\|_2^2}{2\ell^2}\right), \tag{4}$$

where $\ell$ controls the feature similarity decay rate. The covariance matrix $\boldsymbol{\Sigma}$ is constructed using pairwise similarities, which can be written as,

$$\boldsymbol{\Sigma}_{ij} = \sigma^2 S_{ij}, \tag{5}$$

where $\sigma$ governs the perturbation intensity. Stochastic perturbations sampled from $\mathcal{N}(\mathbf{0}, \boldsymbol{\Sigma})$ are then applied to physical properties across particles. However, exact computation of the covariance matrix becomes intractable for large-scale particle systems. We therefore employ the Nyström approximation (Fowlkes et al., 2004), which constructs a low-rank approximation through subset sampling, achieving significant computational efficiency. We thus generate various augmented interaction demonstrations $\{\boldsymbol{X}_t, \boldsymbol{\Phi}, \boldsymbol{a}_t\}_{t=0}^T$ for each scenario, where $\boldsymbol{X}_t = \{\boldsymbol{x}_t^{(i)}\}_{i=1}^N$ represents the object's $N$-particle point cloud at timestep $t$; $\boldsymbol{\Phi} = \{\boldsymbol{\phi}^{(i)}\}_{i=1}^N$ encapsulates per-particle physical properties (e.g., Young's modulus $E_i$ and density $\rho_i$); and $\boldsymbol{a}_t = \{\boldsymbol{a}_t^{(k)}\}_{k=1}^K$ denotes the velocities of $K$ control points interacting with the object at timestep $t$. These data are then used to train a world model.

## 3.3 GNN-BASED WORLD MODELS

Following the construction of a physics-consistent MPM digital twin, we recognize that computational latency is a main limitation to prevent its direct use as a world model. MPM simulations exhibit prohibitive inference times, making them impractical for real-time applications like model-based planning, which require fast responses. To address the issue, we propose to train lightweight and fast GNN-based world models using the augmented synthetic interaction demonstrations generated by the constructed MPM digital twin. The model achieves heterogeneous material-aware dynamics prediction with real-time inference capabilities.

**Model Architecture** The GNN-based world model $f$ can be formulated as:

$$\boldsymbol{X}_{t+1} = f\left(\boldsymbol{X}_{t-h:t}, \boldsymbol{a}_t, \boldsymbol{\Phi}\right), \tag{6}$$

where $t$ denotes the current timestep; $h$ specifies the history window size; $\boldsymbol{a}_t$ represents the applied action(the velocities of control points) at time $t$; and $\boldsymbol{\Phi}$ represents the time-invariant physical properties embedded in GNNs. The GNN construction proceeds through three key steps: First, farthest point sampling (FPS) is applied to the raw point clouds to extract $n$ control particles $\hat{\boldsymbol{X}}_t = \{\hat{\boldsymbol{x}}_t^{(i)}\}_{i=1}^n$ with minimum inter-particle separation $d_v$, while their corresponding physical properties $\boldsymbol{\Phi} = \{\boldsymbol{\phi}^{(i)}\}_{i=1}^N$ are subsampled accordingly to yield $\hat{\boldsymbol{\Phi}} = \{\hat{\boldsymbol{\phi}}^{(i)}\}_{i=1}^n$. Second, control points are incorporated as additional graph vertices. Finally, bidirectional edges $\hat{\boldsymbol{E}}_t$ are established between vertices within connection radius $d_e$.

The GNN architecture integrates three core components: vertex/edge encoders for feature extraction, a multi-step message propagator, and a motion prediction decoder. Designed to ensure translation equivariance, the vertex encoder only processes vertex type indicators (object/controller) encoded as one-hot vectors, control point velocities $\boldsymbol{a}_t$ and time-invariant physical properties $\hat{\boldsymbol{\phi}}^{(i)}$ without vertex positions. Simultaneously, the edge encoder handles history distance between two edge nodes $\{\hat{\boldsymbol{x}}_{t-\tau}^{(i)} - \hat{\boldsymbol{x}}_{t-\tau}^{(j)}\}_{\tau=0}^h$ and a edge type identifier(object-object or object-controller). The output features then undergo $p(p = 7$ in our experiments) iterations of message passing through the propagator's independent MLP blocks with unshared parameters. And the decoder finally generates per-particle 3D motion predictions $\Delta\hat{\boldsymbol{x}}_t^{(i)}$. The entire process can be represented as:

$$\hat{\boldsymbol{X}}_{t+1,pred} = \hat{\boldsymbol{X}}_t + f_\theta(\hat{\boldsymbol{X}}_{t-h:t}, \boldsymbol{a}_t, \hat{\boldsymbol{\Phi}}), \tag{7}$$

where $f_\theta$ denotes the GNN model parameterized by $\theta$.

**Model Training** The generated interaction demonstrations serve as training data for the GNN, with the mean squared error between predicted and ground-truth particle positions constituting the primary loss, *i.e.*,

$$\mathcal{L}_{pred} = \sum_{i=1}^\tau \|\hat{\boldsymbol{X}}_{t+i,pred} - \hat{\boldsymbol{X}}_{t+i}\|^2. \tag{8}$$

The look-forward horizon $\tau$ determines the multi-step prediction window size, balancing accuracy and computational efficiency. Training sequences are initialized at randomly sampled timesteps $t$.

Since our model is trained on ground-truth short-term data pairs, it is susceptible to error accumulation during long-horizon rollouts. To alleviate this issue, we inject noise into historical vertex positions, thereby aligning the training distribution more closely with that encountered during rollout generation. Noise is also introduced to the GNN's physical properties $\hat{\mathbf{\Phi}}$ during training, with the objective of enhancing the stability of subsequent physical property fine-tuning.

**Physical Property Fine-Tuning** Due to the proneness to gradient explosion and training instability when optimizing the physical properties $\hat{\mathbf{\Phi}}$ of MPM particles—a result of employing a large number of simulation substeps per frame—the estimated values of these properties are not yet sufficiently accurate. In contrast, GNNs do not suffer from gradient explosion during training and demonstrate greater stability. Therefore, based on a pre-trained physical property-conditioned GNN, we finetune the physical properties $\hat{\mathbf{\Phi}}$ at the GNN vertices using real-world data while freezing the network parameters $\theta$. This approach enhances the alignment between the GNN and real objects in terms of physical motion characteristics, while also further narrowing the sim-to-real domain gap resulting from training the model solely on synthetic data.

**Action-Conditioned Video Prediction** To achieve action-conditioned video prediction, we additionally optimize object appearance representations using 3DGS (Kerbl et al., 2023). Each object is modeled as a set $\mathcal{G} = \{\mathcal{G}^{(k)}\}_{k=1}^{K}$ of 3D Gaussian kernels, where each $\mathcal{G}^{(k)} = (\boldsymbol{\mu}_k, \mathbf{q}_k, \mathbf{s}_k, \alpha_k, \mathbf{c}_k)$ consists of: center position $\boldsymbol{\mu}_k \in \mathbb{R}^3$, rotation quaternion $\mathbf{q}_k \in \mathbb{SO}(3)$, scaling vector $\mathbf{s}_k \in \mathbb{R}^3$, opacity $\alpha_k \in [0, 1]$, and RGB color coefficients $\mathbf{c}_k \in \mathbb{R}^3$. We optimize the 3D Gaussian Splatting representation solely at $t = 0$. For subsequent timesteps ($t > 0$), the representation is derived via Linear Blend Skinning (LBS) (Sumner et al., 2007), *i.e.*,

$$\mathcal{G}_t = \text{LBS}\left(\mathcal{G}_0, \{\Delta \hat{\boldsymbol{X}}_\tau\}_{\tau=1}^t\right), \tag{9}$$

where LBS interpolates Gaussian motions using the vertex motion field predicted by our GNN. A more detailed description of LBS is provided in Appendix B.4.

## 4 EXPERIMENT

### 4.1 EXPERIMENTAL SETUP

**Implementation Details** Our MPM simulator is built upon PhysGaussian (Xie et al., 2024). We implement additional constitutive models for diverse materials (*e.g.*, anisotropic cloth) and incorporates grasping and pushing capabilities. During MPM simulations, for the majority of simulation scenarios, we set the total simulation duration to 0.04 seconds and the per-frame duration to 0.0001 seconds. This indicates that the state between consecutive frames is advanced through 400 simulation substeps, implicitly satisfying the CFL condition for numerical stability. To mitigate the risk of gradient explosion and ensure training stability during the optimization of physical properties, we computes gradients using only the final 10 simulation substeps, supplemented by gradient clipping. For each scene, we generate 500 interaction demonstration episodes to train the GNN. The frame count for each episode closely matches that of the original real-world data. During GNN training, the number of GNN nodes was maintained at approximately 100 to 150 through FPS downsampling. Meanwhile, the number of message passing steps was set to 7 in our experiments.

**Dataset** We utilize an open-access dataset (Jiang et al., 2025) featuring human interactions with a variety of deformable objects. The dataset consists of 1-10 second videos capturing diverse interactions such as rapid lifting, stretching, pushing, and bimanual squeezing. Each one is with distinct physical properties, including ropes, stuffed animals, cloth, and packages. For each scenario, we partition the whole video into training and test frames with a 7:3 ratio. We only use the training frames exclusively for developing MPM digital twins and fine-tuning the GNN-based world models.

**Evaluation Configurations** Following PhysTwin (Jiang et al., 2025), we employ metrics in both 3D and 2D spaces for evaluation. In 3D space, we utilize the single-direction Chamfer Distance (*i.e.*, CD) and a tracking error (*i.e.*, Track) derived from manually annotated ground-truth points. In 2D space, we evaluate image reconstruction quality using the PSNR, SSIM, and LPIPS (Zhang et al., 2018) metrics, while silhouette alignment is measured by the Intersection over Union (*i.e.*, IoU). To quantify the prediction quality of the model on unseen interactions in the absence of ground truth, we

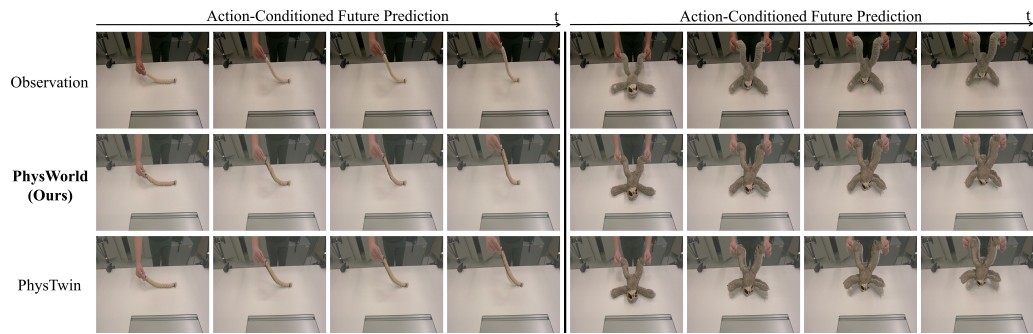

Figure 2: Visual results of action-conditioned future prediction. Our method's predicted positions show closer alignment with ground truth compared to PhysTwin.

Table 1: Quantitative results on action-conditioned future prediction and inference speed (FPS). Best metric values are **bolded**, while second-best ones are underlined.

| Methods | CD↓ | Track↓ | IoU↑ | PSNR↑ | SSIM↑ | LPIPS↓ | FPS↑ |
|---|---|---|---|---|---|---|---|
| Spring-Gaus (ECCV 2024) | 0.062 | 0.094 | 46.4 | 22.488 | 0.924 | 0.113 | 2 |
| GS-Dynamics (CoRL 2024) | 0.041 | 0.070 | 49.8 | 22.540 | 0.924 | 0.097 | 236 |
| PhysTwin (ICCV 2025) | 0.012 | 0.022 | 72.5 | 25.617 | 0.941 | 0.055 | 17 |
| Our MPM | 0.011 | **0.021** | **74.7** | **26.231** | **0.942** | **0.052** | 3 |
| Our GNN w/o Finetuning | 0.012 | 0.025 | 70.0 | 25.345 | 0.939 | 0.059 | **799** |
| PhysWorld(Our GNN w Finetuning) | **0.010** | **0.021** | 73.3 | 25.940 | 0.941 | 0.055 | **799** |

follow DreamPhysics (Huang et al., 2024a) and employ the following metrics from VBench (Huang et al., 2024c): aesthetic quality score, motion smoothness score, and subject consistency score.

**Comparison Configurations** To the best of our knowledge, there is a scarcity of methods for learning object-centric world models from short video clips (under 10 seconds). PhysTwin (Jiang et al., 2025) currently stands as the state-of-the-art approach by employing a spring-mass model as its foundational dynamics model, where the physical properties are optimized through real-world video observations. To provide a more comprehensive evaluation, we include several other approaches, *i.e.*, Spring-Gauss (Zhong et al., 2024) and GS-Dynamics (Zhang et al., 2025b), where Spring-Gauss is a physics-based simulation method also based on spring-mass models and GS-Dynamics is a learning-based approach employing a GNN-based model to learn dynamics from real videos.

## 4.2 EXPERIMENTAL RESULTS

**Action-Conditioned Future Prediction** The qualitative results in Fig. 2 and quantitative results in Table 1 for the action-conditioned future prediction task demonstrate the superior performance of our proposed methods. Our MPM method, enhanced with optimized physical properties, surpasses other approaches across most evaluated metrics, which underscores MPM's ability to simulate a wide variety of deformable objects with high fidelity. Furthermore, our finetuned GNN achieves competitive results, outperforming the baseline model PhysTwin in the majority of benchmarks and achieving the lowest CD and Track loss with the fastest inference speed. This highlights the accuracy and effectiveness of PhysWorld in both predicting motion and generating realistic images.

**Inference Time Comparison** We perform the inference speed comparisons with an NVIDIA GeForce RTX 4060 Ti (16GB) GPU in the *double_lift_cloth_3* scene, which consists of 118 frames (a relatively long sequence) and contains 171,602 3DGS kernels. The results are shown in Table 1. The results demonstrate our PhysWorld maintains competitive accuracy while achieving fast inference speed. Spring-Gaus exhibits limitations in both prediction accuracy and inference speed. Although GS-Dynamics is observed to have a fast inference speed, it suffers from low predictive accuracy. The recent state-of-the-art PhysTwin lags behind in both speed ($47\times$ slower) and prediction accuracy. MPM achieves slightly better prediction accuracy but at a prohibitive computational cost, running about $400\times$ slower than PhysWorld. The real-time inference capability of PhysWorld

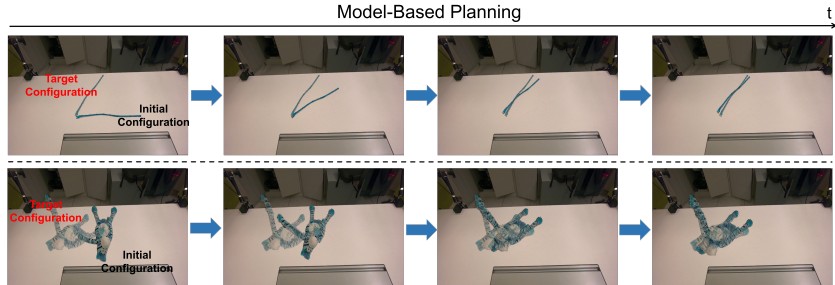

Figure 3: Generalization to unseen interactions. As representative examples, we consider two unseen interaction scenarios: lifting a pushed rope and rotating a lifted sloth. The results show that PhysWorld generates physically plausible predictions, while PhysTwin suffers from artifacts such as fracture-like rope distortions and unnatural foot folding.

Figure 4: Examples of model-based planning. With MPPI control, the rope and the zebra doll are transferred from the initial configurations to the target ones.

Table 2: Quantitative results on unseen interactions.

| Methods | Aesthetic Quality↑ | Motion Smoothness↑ | Subject Consistency↑ |
|---|---|---|---|
| Phystwin | 0.4315 | 0.9971 | 0.9155 |
| PhysWorld (Ours) | **0.4440** | **0.9973** | **0.9312** |

enables its deployment in computationally demanding scenarios, such as model-based robotic planning, interactive simulation systems, and real-time optimization tasks.

**Generalization to Unseen Interactions** We further compare PhysWorld's generalization performance on unseen interactions. Using *single_push_rope* and *double_lift_sloth* as test cases, we apply randomized action sequences to objects. Visual results in Fig. 3 show that PhysWorld consistently generates high-fidelity physical predictions, significantly outperforming PhysTwin in dynamic accuracy and deformation realism. The quantitative results in Table 2 also indicate that PhysWorld generates more plausible predictions when confronted with unseen interactions.

**Model-Based Planning** PhysWorld enables real-time trajectory optimization for model-based robotic planning, as shown in the case studies in Fig. 4. Given initial and target configurations of the deformable objects, we utilize a Model-Predictive Path Integral (MPPI) (Williams et al., 2017) planning framework, which successfully generates controlling trajectories to guide the objects towards their target configurations.

## 4.3 ABLATION STUDY

**Effect of Global-to-Local Physical Property Optimization** Here we conduct an ablation study on optimization strategies of physical properties within MPM. As summarized in Table 3, experimental results demonstrate that global-only optimization fails to capture localized material variations, while local optimization exhibits convergence challenges. Our global-to-local approach significantly enhances optimization stability in the second stage by initializing local properties with global property priors, yielding superior overall performance.

**Effect of VMP-Gen** The ablation studies (Table 4) demonstrate that the proposed VMP-Gen modules significantly enhance the trained GNN's prediction accuracy and robustness. This improvement

Table 3: Effect of different physical property optimization strategies for MPM.

| Strategy | CD↓ | Track↓ | IoU↑ | PSNR↑ | SSIM↑ | LPIPS↓ |
|---|---|---|---|---|---|---|
| Global | 0.012 | 0.024 | 72.4 | 25.854 | 0.940 | 0.055 |
| Local | 0.016 | 0.032 | 66.5 | 24.901 | 0.935 | 0.066 |
| Global-to-Local (Ours) | **0.010** | **0.021** | **74.7** | **26.231** | **0.942** | **0.052** |

Table 4: Effect of different motion patterns.

| Motion Pattern | CD↓ | Track↓ | IoU↑ | PSNR↑ | SSIM↑ | LPIPS↓ |
|---|---|---|---|---|---|---|
| Uniform and linear | 0.0114 | 0.0175 | 76.95 | 24.475 | 0.920 | **0.067** |
| VMP-Gen (Ours) | **0.0100** | **0.0154** | **78.66** | **24.666** | **0.921** | **0.067** |

Table 5: Effect of different physical property perturbation strategies.

| Perturbation | CD↓ | Track↓ | IoU↑ | PSNR↑ | SSIM↑ | LPIPS↓ |
|---|---|---|---|---|---|---|
| × | 0.0111 | 0.0179 | 75.84 | 23.984 | 0.919 | 0.070 |
| Random | 0.0153 | 0.0216 | 70.19 | 23.073 | 0.915 | 0.082 |
| Uniform | 0.0147 | 0.0258 | 72.00 | 23.101 | 0.914 | 0.079 |
| $P^3$-Pert (Ours) | **0.0100** | **0.0154** | **78.66** | **24.666** | **0.921** | **0.067** |

Table 6: Performance comparison with a GNN directly trained on real data.

| Methods | CD↓ | Track↓ | IoU↑ | PSNR↑ | SSIM↑ | LPIPS↓ |
|---|---|---|---|---|---|---|
| GNN (Directly trained on real data) | 0.0530 | 0.0802 | 41.32 | 19.925 | 0.881 | 0.132 |
| PhysWorld (Ours) | **0.0100** | **0.0154** | **78.66** | **24.666** | **0.921** | **0.067** |

is achieved by increasing motion diversity, enabling the model to capture a wider range of patterns beyond simple uniform linear motion.

**Effect of $P^3$-Pert** Ablation studies demonstrate the effectiveness of the proposed $P^3$-Pert modules, with quantitative results detailed in Table 5. The key advantage of $P^3$-Pert lies in its generation of physically realistic part-aware property variations. As evidenced by the results, our method surpasses the random and uniform perturbation baselines (e.g., AdaptiGraph (Zhang et al., 2024a)) by producing more plausible and diverse physical property distributions.

**Training GNN Directly on Real Data** To evaluate the role of synthetic data in GNN training, we compared PhysWorld against traditional real-data-only training. Results on the *double_lift_sloth* task in Table 6 indicate that real-data scarcity caused severe overfitting in GNNs, which, as a result, led to compromised performance in future state prediction due to poor generalization. In contrast, incorporating various synthetic data used in PhysWorld significantly improved prediction accuracy and model robustness.

## 5 CONCLUSION

In this work, we introduce PhysWorld, a novel framework that constructs accurate and efficient world models for deformable objects through physics-aware demonstration synthesis from real-world videos. It first establishes a MPM-based digital twin and then generates extensive augmented interaction demonstrations via strategic perturbations of physical parameters, control trajectories, and velocity profiles. We employ the generated data to train a real-time GNN world model that learns spatially heterogeneous physical properties, with real video data subsequently refining the model's parameters. The world model supports diverse downstream applications such as model-based planning and robotic manipulation. Extensive experiments show that PhysWorld outperforms state-of-the-art methods.

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

# APPENDIX

The content of this supplementary material includes:

- Details about Material Point Method (MPM) in Sec. A;
- Details of our method in Sec. B;
- Additional ablation studies in Sec. C.
- Additional analysis and results in Sec. D.

## A MATERIAL POINT METHOD (MPM)

The Material Point Method (MPM) (Jiang et al., 2016; Stomakhin et al., 2013; Bardenhagen et al., 2000) is particularly powerful for simulating complex nonlinear mechanics in highly deformable objects, including soft materials (e.g. rubber, foam, biological tissues), granular media (e.g., soil, snow), and fluids undergoing extreme plastic flows. Its core advantage lies in its hybrid Lagrangian-Eulerian formulation: Lagrangian material points inherently track mass, momentum, complete deformation history (captured via the deformation gradient tensor), and evolving material state (stress, damage) through arbitrary motion and topological changes, while a background computational grid solves momentum equations on-the-fly automatically avoiding mesh entanglement issues inherent to traditional mesh-based methods like FEM. This unique synergy enables MPM to natively handle extreme deformations, automatic material separation/fracture, and history-dependent constitutive behavior without remeshing or special failure criteria.

MPM operates through three key phases: Particle-to-Grid (P2G) Transfer, Grid Operations(GridOp), and Grid-to-Particle (G2P) Transfer.

### A.1 PARTICLE-TO-GRID (P2G) TRANSFER.

During the Particle-to-Grid (P2G) phase, material points transfer their mass and momentum to grid nodes via interpolation functions. Using a fixed timestep $\Delta t$ such that $t^n = n\Delta t$, the mass $m_i^n$ at grid node $i$ at timestep $n$ is computed as:

$$m_i^n = \sum_p w_{ip}^n m_p \tag{10}$$

where $m_p$ is the mass of material point $p$, and $w_{ip}^n$ denotes the B-spline interpolation weight between material point $p$ and grid node $i$. Similarly, grid momentum is calculated through:

$$m_i^n \mathbf{v}_i^n = \sum_p w_{ip}^n m_p \left( \mathbf{v}_p^n + \mathbf{C}_p^n (\mathbf{x}_i - \mathbf{x}_p^n) \right) \tag{11}$$

where $\mathbf{v}_p^n$ is the material point's velocity, $\mathbf{C}_p^n$ represents its affine momentum matrix, and $\mathbf{x}_i$, $\mathbf{x}_p^n$ denote grid node and material point positions respectively.

### A.2 GRID OPERATIONS (GRIDOP).

The Grid Operations (GridOp) phase constitutes the computational core of MPM, where the governing equations of continuum mechanics are solved on the background grid. Following Particle-to-Grid (P2G) transfer, this phase performs two critical operations:

First, grid velocities are updated by resolving both internal stresses and external forces:

$$\hat{\mathbf{v}}_i^{n+1} = \mathbf{v}_i^n + \Delta t \left( \frac{1}{m_i^n} \sum_p \boldsymbol{\sigma}_p^n \nabla w_{ip}^n V_p^0 + \mathbf{g} \right) \tag{12}$$

where $\boldsymbol{\sigma}_p^n$ denotes the Cauchy stress tensor at material point $p$ (derived from its deformation state), $V_p^0$ is the material point's initial volume, and $\mathbf{g}$ represents gravitational acceleration. This explicit integration accounts for internal forces and external body forces.

Subsequently, boundary conditions are enforced through velocity modifications:

$$\mathbf{v}_i^{n+1} = \mathcal{P}_{\partial\Omega}\left(\hat{\mathbf{v}}_i^{n+1}\right) \tag{13}$$

where $\mathcal{P}_{\partial\Omega}$ is a projection operator imposing constraints at domain boundaries $\partial\Omega$. For Dirichlet boundaries (e.g., collider surfaces), signed distance fields (SDF) modify velocities to satisfy prescribed kinematic conditions.

### A.3 GRID-TO-PARTICLE (G2P) TRANSFER.

The Grid-to-Particle (G2P) phase transfers the updated grid kinematics back to material points, completing the time step. Material point velocities are first reconstructed through grid velocity interpolation:

$$\mathbf{v}_p^{n+1} = \sum_i \mathbf{v}_i^{n+1} w_{ip}^n \tag{14}$$

where $w_{ip}^n$ is the consistent B-spline interpolation weight. Material point positions are then updated via explicit advection:

$$\mathbf{x}_p^{n+1} = \mathbf{x}_p^n + \Delta t \mathbf{v}_p^{n+1} \tag{15}$$

To preserve local deformation characteristics and reduce numerical diffusion, the affine velocity field matrix is reconstructed:

$$\mathbf{C}_p^{n+1} = \frac{4}{(\Delta x)^2} \sum_i w_{ip}^n \mathbf{v}_i^{n+1} (\mathbf{x}_i - \mathbf{x}_p^n)^T \tag{16}$$

where $\Delta x$ is the grid spacing. Finally, the deformation gradient tensor undergoes incremental updating to track finite-strain kinematics:

$$\boldsymbol{F}_p^{n+1} = \left(\boldsymbol{I} + \Delta t \mathbf{C}_p^{n+1}\right) \boldsymbol{F}_p^n \tag{17}$$

This deformation gradient update enables history-dependent constitutive evaluation in the subsequent time step, closing the MPM computational loop.

## B METHOD DETAILS

### B.1 VLM-ASSISTED CONSTITUTIVE MODEL SELECTION

Our framework leverages Qwen3 Yang et al. (2025)'s multimodal capabilities to automate constitutive model selection for MPM simulations. By processing object manipulation videos, we extract deformation sequences through optical flow variance detection. Qwen3 then analyzes the material responses observed in these videos, matching the underlying physics descriptors against our curated library of constitutive laws following Jiang et al. (2016); Xie et al. (2024). This library includes plasticity models for describing irreversible deformation and elasticity models for computing Kirchhoff stress:

1. **Plasticity Models:** Encompass Drucker-Prager for pressure-sensitive yielding in geomaterials, Von Mises for $J^2$ plasticity in metals, and St. Venant-Kirchhoff for finite-strain elasticity preceding yield or purely elastic behavior.

2. **Elasticity Models:** Feature Fixed Co-rotated (FCR) for robustness under large rotations, Neo-Hookean with volumetric coupling, St. Venant-Kirchhoff for direct mapping to Green strain, Drucker-Prager Elastic with pressure-dependent stiffness and Anisotropic Hyperelasticity for cloth.

Qwen3 performs model selection through structured prompting. The system receives the following instruction:

```
Analyze the video of object-hand interaction and recommend
optimal MPM constitutive models from the following library:
  • Elastic Models:  FCR, Neo-Hookean, St. Venant-Kirchhoff,
    Anisotropic Hyperelastic
```

```
    • Plastic Models:  Von Mises, Drucker-Prager,
      St. Venant-Kirchhoff, Purely Elastic

  Provide recommendations in the format:
  Elastic:  [model] | Plastic:  [model] | Reason:
  [justification]
```

When video evidence demonstrates matching physical behaviors, Qwen3 outputs the best-matched constitutive models from our library. We evaluated the accuracy of Qwen3 in predicting plasticity models and elasticity models across 22 scenarios. The experimental results demonstrate that Qwen3 achieved perfect prediction of $100\%$ accuracy for both plasticity models and elasticity models in all 22 scenarios. Specifically, elasticity models for 8 cloth-type objects were consistently predicted as anisotropic hyperelastic, while Neo-Hookean was assigned to the other 14 objects. Additionally, the plasticity models for all objects were correctly identified as purely elastic.

## B.2 PHYSICAL PROPERTY OPTIMIZATION

We implement a global-to-local optimization strategy to calibrate the physical properties of an MPM-based digital twin through two sequential computational stages. The initial global homogenization stage optimizes fundamental domain-wide parameters—including Young's modulus $E$, friction coefficient $\mu$, Poisson's ratio $\nu$, yield stress $\sigma_y$, and mass density $\rho$—to establish stable dynamic behavior. This provides robust initialization for the subsequent heterogeneous refinement stage, which selectively optimizes only $E$, $\mu$, and $\rho$ with spatial variation to ensure convergence while avoiding over-parameterization. Given the predicted point positions $\hat{X}_t = \{\hat{x}_t^{(i)}\}_{i=1}^N$ and the ground truth $X_t = \{x_t^{(i)}\}_{i=1}^N$ at time $t$, both stages employ identical composite loss metrics combining geometric matching and kinematic consistency:

$$\mathcal{L}_{\text{CD}}(X_t, \hat{X}_t) = \frac{1}{|X_t|} \sum_{x \in X_t} \min_{y \in \hat{X}_t} \|x - y\|$$
$$+ \frac{1}{|\hat{X}_t|} \sum_{y \in \hat{X}_t} \min_{x \in X_t} \|y - x\| \tag{18}$$

$$\mathcal{L}_{\text{track}}(X_t, \hat{X}_t) = \frac{1}{3|\mathcal{V}|} \sum_{i \in \mathcal{V}} \sum_{c \in \{x,y,z\}} \ell_{smooth\_l1}(\|\hat{x}_{t,c}^{(i)} - x_{t,c}^{(i)}\|) \tag{19}$$

where $\mathcal{V}$ represents point indices, and the smooth $\ell_1$ component is defined as:

$$\ell_{smooth\_l1}(d) = \begin{cases} \frac{1}{2}d^2 & \text{if } d < 1 \\ d - 0.5 & \text{otherwise} \end{cases} \tag{20}$$

The total optimization objective combines these metrics through weighted summation:

$$\mathcal{L}_{\text{total}} = \lambda_1 \mathcal{L}_{\text{CD}} + \lambda_2 \mathcal{L}_{\text{track}} \tag{21}$$

with experimentally configured weights $\lambda_1 = 1.0$ and $\lambda_2 = 0.1$.

## B.3 FINE-TUNING PHYSICAL PROPERTY FOR GNN-BASED WORLD MODELS

When training GNN-based world models using generated demonstrations, we concurrently input the physical properties of objects from each demonstration into the Graph Neural Network (GNN). This process yields a physics-informed GNN capable of dynamically adapting its physical dynamics predictions according to varying input physical features. To align the simulated physical properties with real-world objects for consistent kinematic behavior, we subsequently fine-tune these properties using tracking points extracted from real videos. During fine-tuning, we freeze the GNN model parameters $\theta$ while optimizing only the physical properties $\hat{\mathbf{\Phi}}$ associated with each graph vertex. Due to the discrepancy between the number of GNN vertices and ground truth tracking points, we employ Linear Blend Skinning (LBS) to interpolate positions for all target track points from the GNN vertex outputs. The optimization utilizes a composite loss function combining unidirectional Chamfer distance and Mean Squared Error (MSE) Loss.

### B.4 3D GAUSSIAN UPDATE THROUGH LBS

To predict the appearance of deformable objects at time $t$ represented by a set of 3D Gaussian kernels $\mathcal{G}_{t+1}$, we first compute the 6-DoF transformation for each vertex $\hat{\mu}_i^t \in \hat{X}_t$, consisting of a translation $T_i^t$ and rotation $R_i^t$. The translation is directly obtained from the vertex displacement:

$$T_i^t = \hat{\mu}_i^{t+1} - \hat{\mu}_i^t = \hat{x}_{t+1}^i - \hat{x}_t^i. \tag{22}$$

For 3D rotation, we estimate a rigid local rotation $R_i^t$ for each vertex by minimizing the motion discrepancy of its neighborhood $\mathcal{N}(i)$ between $t$ and $t+1$:

$$R_i^t = \arg\min_{R \in \mathrm{SO}(3)} \sum_{j \in \mathcal{N}(i)} \left\| R(\hat{\mu}_j^t - \hat{\mu}_i^t) - (\hat{\mu}_j^{t+1} - \hat{\mu}_i^{t+1}) \right\|^2, \tag{23}$$

We then transform Gaussian kernels via Linear Blend Skinning (LBS) Huang et al. (2024b); Sumner et al. (2007); Zhang et al. (2024b); Jiang et al. (2025) by interpolating transformations from their neighboring GNN vertices. The 3D position and rotation of each Gaussian can be computed by:

$$\mu_j^{t+1} = \sum_{k \in \mathcal{N}(j)} w_{jk}^t (R_k^t (\mu_j^t - \hat{\mu}_k^t) + \hat{\mu}_k^t + T_k^t) \tag{24}$$

$$q_j^{t+1} = \left( \sum_{k \in \mathcal{N}(j)} w_{jk}^t r_k^t \right) \otimes q_j^t, \tag{25}$$

where $R_k^t \in \mathbb{R}^{3\times3}$ and $r_k^t \in \mathbb{R}^4$ respectively denote the rotation matrix and quaternion of vertex $k$; $\otimes$ represents the quaternion multiplication; $\mathcal{N}(j)$ indicates $K$-nearest GNN vertices to Gaussian kernel $j$; and $w_{jk}^t$ defines the interpolation weights between a Gaussian $\mu_j^t$ and a corresponding GNN vertex $\hat{\mu}_k^t$, which is inversely proportional to their Euclidean distance:

$$w_{jk}^t = \frac{\|\mu_j^t - \hat{\mu}_k\|^{-1}}{\sum_{k \in \mathcal{N}(j)} \|\mu_j^t - \hat{\mu}_k\|^{-1}} \tag{26}$$

to ensure that spatially closer Gaussian-vertex pairs receive higher weighting influence. With the updated 3D Gaussian parameters, we render the deformable objects to obtain their appearance at $t+1$.

## C ADDITIONAL ABLATION STUDIES

### C.1 BACKBONE ARCHITECTURE ABLATION

To examine the effect of backbone choice on deformable object dynamics learning, we consider two alternative architectures in place of our GNN:

- **MLP backbone**. We implement a point-based MLP model following PointMLP (Ma et al., 2022).
- **Transformer backbone**. We implement a Transformer model following PointTransformerV3 (Wu et al., 2024c).

All variants are trained on the *double_lift_sloth* scenario under the same data, losses, and optimization protocol as PhysWorld. We then evaluate their prediction accuracy and inference speed using the metrics in the main paper. The results are demonstrated in Table. 7. The results show that the GNN backbone achieves the best prediction accuracy across all metrics, and is substantially faster at inference. We attribute this advantage to the inductive bias of GNNs for deformable dynamics: the radius-graph construction and multi-step message passing explicitly model sparse, local particle–particle and particle–tool interactions, which aligns well with how forces propagate in deformable media. In contrast, MLPs must learn such interactions implicitly from point features, while Transformers incur dense attention overhead without explicit locality priors. These properties make GNNs a practical and effective choice for real-time, material-aware deformable world modeling.

Table 7: Comparison of different backbone architectures for action-conditioned future prediction and inference speed (FPS).

| Methods | CD↓ | Track↓ | IoU↑ | PSNR↑ | SSIM↑ | LPIPS↓ | FPS↑ |
|---|---|---|---|---|---|---|---|
| Transformer | 0.0497 | 0.0656 | 42.43 | 20.291 | 0.891 | 0.133 | 56 |
| MLP | 0.0213 | 0.0343 | 70.13 | 23.206 | 0.914 | 0.084 | 107 |
| Our GNN | **0.0100** | **0.0154** | **78.66** | **24.666** | **0.921** | **0.067** | **784** |

Table 8: GNN distillation results using PhysTwin as the teacher simulator. Incorporating VMP-Gen and $P^3$-Pert improves performance, while MPM-based PhysWorld achieves the best overall results.

| Method | VMP-Gen & $P^3$-Pert | CD↓ | Track↓ | IoU↑ | PSNR↑ | SSIM↑ | LPIPS↓ |
|---|---|---|---|---|---|---|---|
| PhysTwin + GNN | × | 0.0296 | 0.0679 | 55.66 | 21.220 | 0.895 | 0.121 |
| PhysTwin + GNN | ✓ | 0.0187 | 0.0320 | 67.87 | 22.686 | 0.911 | 0.082 |
| MPM + GNN (Ours) | ✓ | **0.0100** | **0.0154** | **78.66** | **24.666** | **0.921** | **0.067** |

## C.2 PHYSTWIN + GNN DISTILLATION EXPERIMENTS

To further examine the role of our data-generation modules (VMP-Gen and $P^3$-Pert) and to contextualize the benefits of using an MPM-based digital twin as the teacher model, we conducted an additional set of distillation experiments in which the GNN is trained using PhysTwin as the supervisory simulator. For the *double_lift_sloth* scene, we replace the MPM simulator with PhysTwin's spring–mass model and generate synthesized demonstrations using the same data-generation protocol as PhysWorld.

We consider two settings: (1) training on synthesized demonstrations without VMP-Gen or $P^3$-Pert, and (2) training with both modules enabled. In all cases, the GNN architecture, losses, and optimization schedule are identical to those used in the main paper. Table 8 summarizes the performance of the distilled GNN. Incorporating VMP-Gen and $P^3$-Pert improves the quality of PhysTwin-generated training data, leading to a stronger GNN even when PhysTwin serves as the teacher. When compared to PhysWorld, the GNN distilled from MPM still achieves the highest accuracy across all metrics. These results show that VMP-Gen and $P^3$-Pert substantially enhance the effectiveness of GNN distillation even when the supervisory simulator is PhysTwin and the MPM-based digital twin provides higher-fidelity training signals, leading to the best-performing distilled GNN in our framework.

## D ADDITIONAL ANALYSIS AND RESULTS

### D.1 ADDITIONAL QUANTITATIVE RESULTS FOR MPPI-BASED PLANNING

We use an MPPI-based planner, where at each planning step we sample 2000 candidate trajectories, roll them out with our learned dynamics model, compute the distance between the terminal state of each trajectory and the target configuration, assign higher weights to trajectories with smaller terminal errors, and obtain the control by taking a weighted average over all trajectories; only the first action of this averaged control sequence is executed, after which we re-sample and repeat the procedure in a receding-horizon fashion. The results in Fig. 5 compare our method(GNN) with PhysTwin and MPM in two deformable object relocating tasks by plotting the distance error between the objects's current and target positions as a function of the planning steps. Fig. 6 further compares these three methods in terms of success-rate curves under different error thresholds. A comparison of the wall-clock time required for 10-step MPPI planning is also illustrated in Table. 9, where each step samples 2000 trajectories. Thanks to the GNN's ability to perform parallel inference over multiple trajectories, our method requires significantly less planning time while still achieving strong control performance.

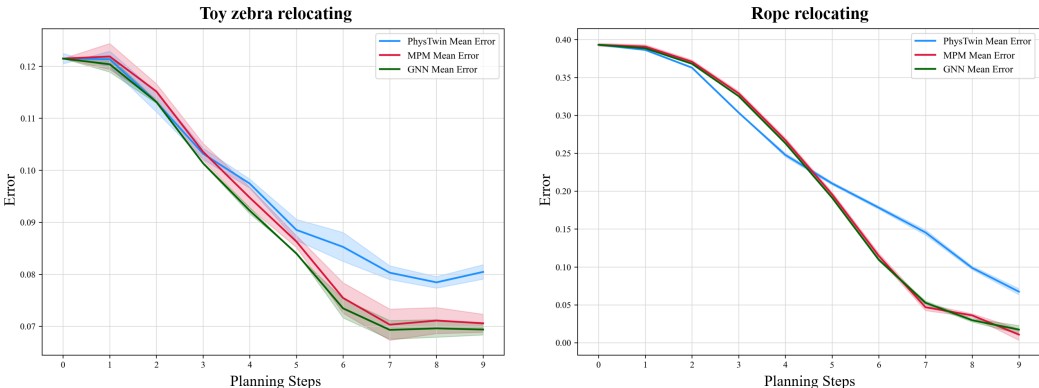

Figure 5: Mean error curves over planning steps for different methods (PhysTwin, our MPM, and our GNN) on the toy-zebra relocation (left) and rope relocation (right) tasks.

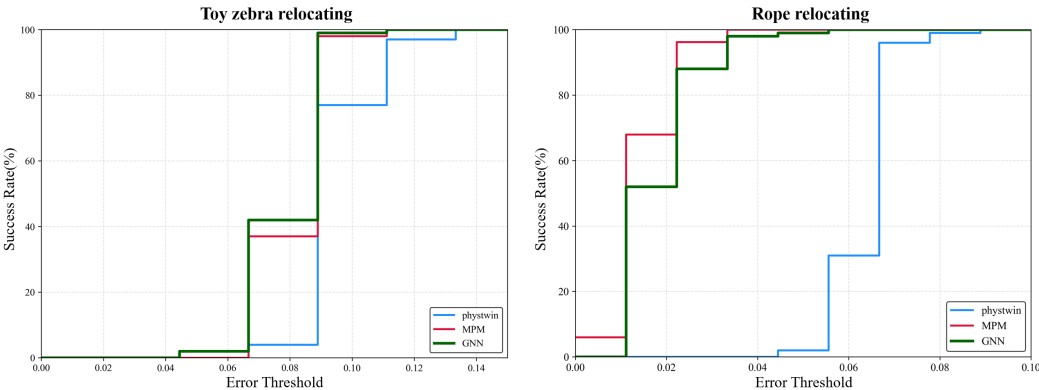

Figure 6: Success rate curves with respect to different error thresholds for different methods (PhysTwin, our MPM, and our GNN) on the toy-zebra relocation (left) and rope relocation (right) tasks.

### D.2 PHYSICAL REALISM OF SYNTHESIZED DEMONSTRATIONS

To assess the physical realism of our synthesized demonstrations, we adopt the evaluation protocol of PhyGenBench (Meng et al., 2024). This benchmark measures the consistency of object motion with 27 physical laws, including those related to gravity, friction, elastic forces, and other basic dynamics. We apply these metrics to both our generated demonstrations and the real ground-truth videos. Our synthesized demonstrations achieve an average Realism score of $1.6010$, which is very close to the real videos' score of $1.6087$, indicating that the generated object motions are highly physically realistic.

### D.3 ROBUSTNESS OF VLM-ASSISTED CONSTITUTIVE MODEL SELECTION

To further evaluate the robustness of our VLM-based constitutive model selection, we collected additional videos from the Internet showing hand/robot-arm interactions with deformable objects(as shown in Fig. 7) and asked Qwen3 to choose the most suitable constitutive model for the manipulated object from our material library, containing 4 elastic models and 4 plastic models. We then compared Qwen3's predictions with expert annotations and observed consistently high agreement (100%) accuracy on this extended set. This suggests that, with a reasonably designed mate-

Table 9: Per-step planning time comparison for different methods.

| Methods | PhysTwin | MPM | GNN(Ours) |
|---|---|---|---|
| Planning time per step (s) | 998 | 4687 | **22** |

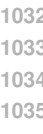
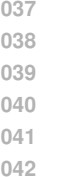
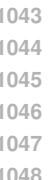
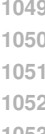

Figure 7: Examples of additional collected video clips showing hand/robot-arm interactions with deformable objects.

rial library, constitutive-model selection using a modern VLM is a manageable subproblem in our pipeline.

We also note that picking a "second-best" model does not necessarily break the system. In practice, some objects can be described reasonably well by more than one constitutive law. For instance, many cloth-like objects are best matched by an anisotropic hyperelastic model, but a simpler Neo-Hookean elastic model can still yield plausible behavior for thicker fabrics. To test robustness, we conducted this study on the *single_lift_cloth_1* scene, where the theoretically optimal constitutive model is Anisotropic Hyperelastic. We intentionally replaced it with a suboptimal model-Neo-Hookean and reran the pipeline. The results in Table. 10 show that our framework can re-optimize the physical parameters under this suboptimal choice and still recover accurate dynamics, indicating tolerance to occasional non-optimal model selection.

## D.4 GNN TRAINING SENSITIVITY

To assess the robustness of our GNN training procedure, we ran additional experiments with three different random seeds in Fig. 8 (leading to different parameter initializations) and varied the batch size and learning rate in Fig. 9 and Fig. 10 respectively. We found that changing random seeds or batch size has negligible impact on final performance, and the training loss curves consistently converge to similar values. The learning rate is also stable within a reasonable range; however, when it is set too small, convergence becomes noticeably slower (though it still reaches a comparable minimum). This indicates that PhysWorld is not overly sensitive to initialization or most hyperparameters.

Table 10: Robustness to suboptimal VLM constitutive-model selection: comparison of PhysTwin and our pipeline using the optimal vs. an intentionally suboptimal constitutive model.

| Methods | CD↓ | Track↓ | IoU↑ | PSNR↑ | SSIM↑ | LPIPS↓ |
|---|---|---|---|---|---|---|
| PhysTwin | 0.0221 | 0.0286 | 55.75 | 23.919 | 0.958 | 0.062 |
| Ours (Suboptimal material) | 0.0193 | 0.0339 | 62.15 | 26.512 | 0.968 | 0.046 |
| Ours (Optimal material) | **0.0186** | **0.0241** | **75.54** | **27.700** | **0.974** | **0.039** |

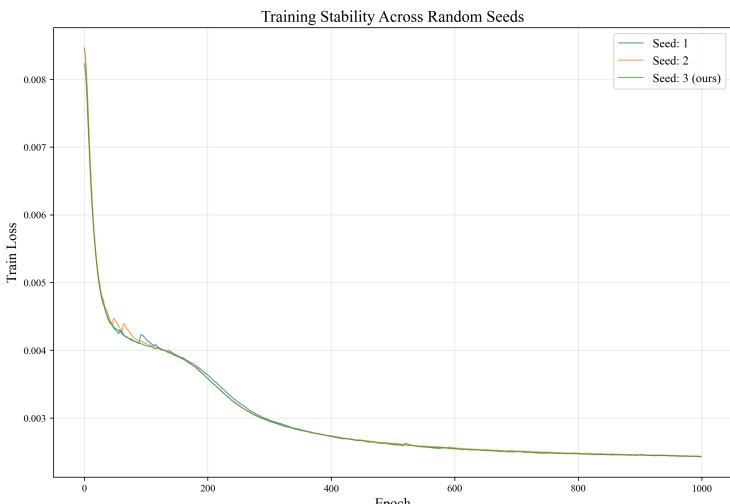

Figure 8: Training loss curves over epochs with different random seeds.

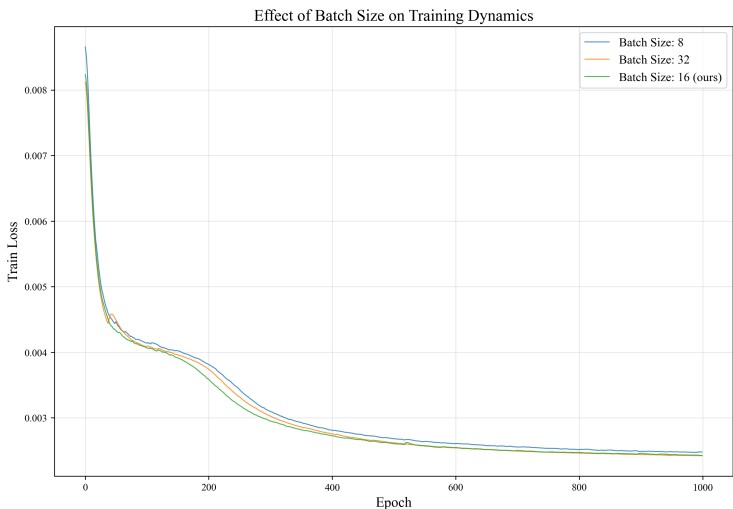

Figure 9: Training loss curves over epochs with different batch sizes.

## D.5 OVERALL COMPUTATIONAL COST

For completeness, we report the end-to-end computational cost of PhysWorld for a representative scene, *double_lift_sloth*, evaluated on a single RTX 4060 Ti (16 GB) GPU. The pipeline consists of four stages: (1) MPM-based digital twin optimization, (2) demonstration synthesis, (3) GNN training, and (4) GNN fine-tuning on real videos. On average, the end-to-end process requires a total of **8.89 h** per object, with the stage-wise breakdown summarized in Table 11. While this one-

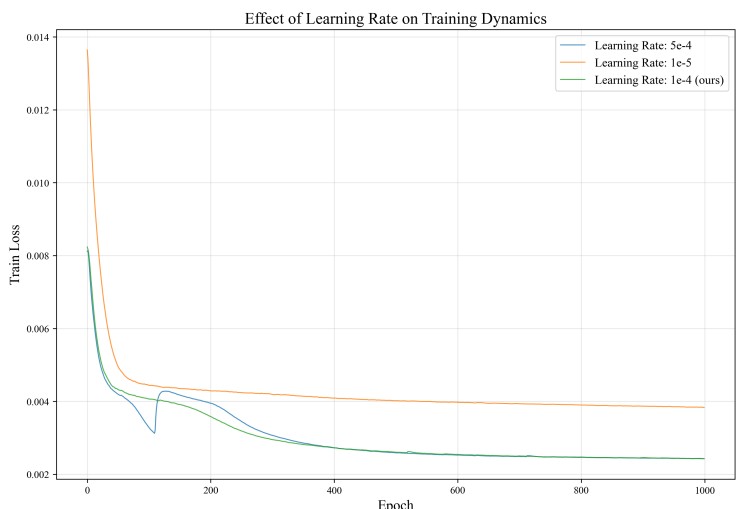

Figure 10: Training loss curves over epochs with different learning rates.

Table 11: Overall computational cost.

| Stage | Duration (h) | Share of total |
|---|---|---|
| MPM optimization | 0.38 | 4.3% |
| Demonstration synthesis | 3.00 | 33.7% |
| GNN training | 5.48 | 61.7% |
| GNN fine-tuning | 0.03 | 0.3% |
| **Total** | **8.89** | **100%** |

time preparation cost is non-trivial, it yields a fast learned world model that runs orders of magnitude faster than the underlying MPM simulator and can be directly used for downstream tasks such as model-based planning and control, where high inference speed is crucial.

### D.6  EVALUATION ON ELASTO-PLASTIC OBJECTS FROM ROBOCRAFT

To assess the advantage of using an MPM-based digital twin on materials beyond the elastic regime, we conducted additional experiments on elasto-plastic objects from RoboCraft (Shi et al., 2024). These objects undergo large and partially irreversible deformations, which are challenging to capture with spring-mass dynamics. For each RoboCraft sequence, we calibrate physical parameters independently for (1) an MPM simulator and (2) PhysTwin's spring-mass simulator, using the same optimization procedure as in our main pipeline. We then roll out both calibrated simulators under the recorded actions and compare their predicted object trajectories against ground-truth point clouds. Table 12 reports quantitative errors. The calibrated MPM digital twin consistently achieves lower Chamfer, EMD, and Hausdorff losses than PhysTwin, indicating more faithful reproduction of elasto-plastic dynamics, especially under large and irreversible deformations.

Furthermore, as illustrated in Fig. 11, when the two control points push inward on the elasto-plastic object and then move back, the behaviors of the two simulators differ markedly. PhysTwin, built on a spring-mass formulation, tends to enforce elastic recovery, causing the object to automatically return toward its original shape after the push. In contrast, the calibrated MPM digital twin accurately predicts the persistent, plastic deformation induced by this interaction, producing trajectories that remain consistent with the irreversible shape changes observed in the real sequence.

Table 12: Comparison of calibrated MPM and PhysTwin on elasto-plastic objects from RoboCraft.

| Method | Chamfer Loss↓ | EMD Loss↓ | Hausdorff Loss↓ |
|---|---|---|---|
| PhysTwin | 0.0190 | 0.0320 | 0.1701 |
| Our MPM | **0.0175** | **0.0276** | **0.0963** |

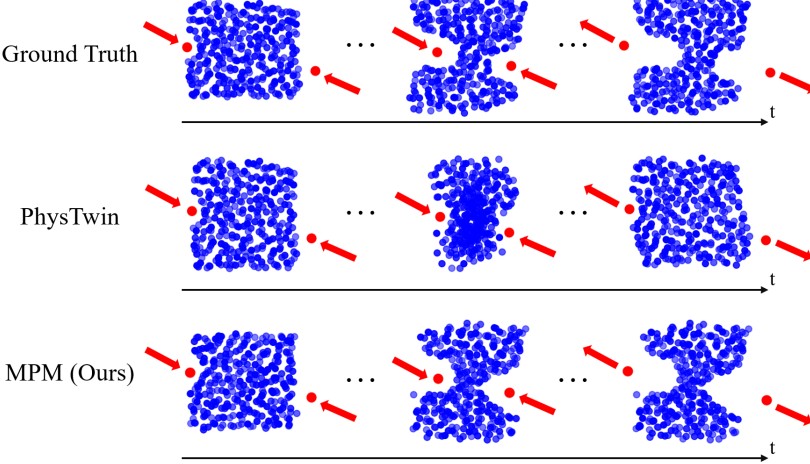

Figure 11: Qualitative results of PhysTwin and calibrated MPM on an elasto-plastic RoboCraft object. When two control points push inward and then retract, PhysTwin's spring–mass simulator exhibits elastic recovery toward the original shape, whereas MPM captures the persistent plastic deformation, matching the irreversible shape changes observed in the real sequence.

## D.7 ADDITIONAL QUALITATIVE RESULTS

We present additional qualitative results of action-conditioned future prediction in Fig. 12, demonstrating the superior performance of our method compared to the SOTA method PhysTwin Jiang et al. (2025).

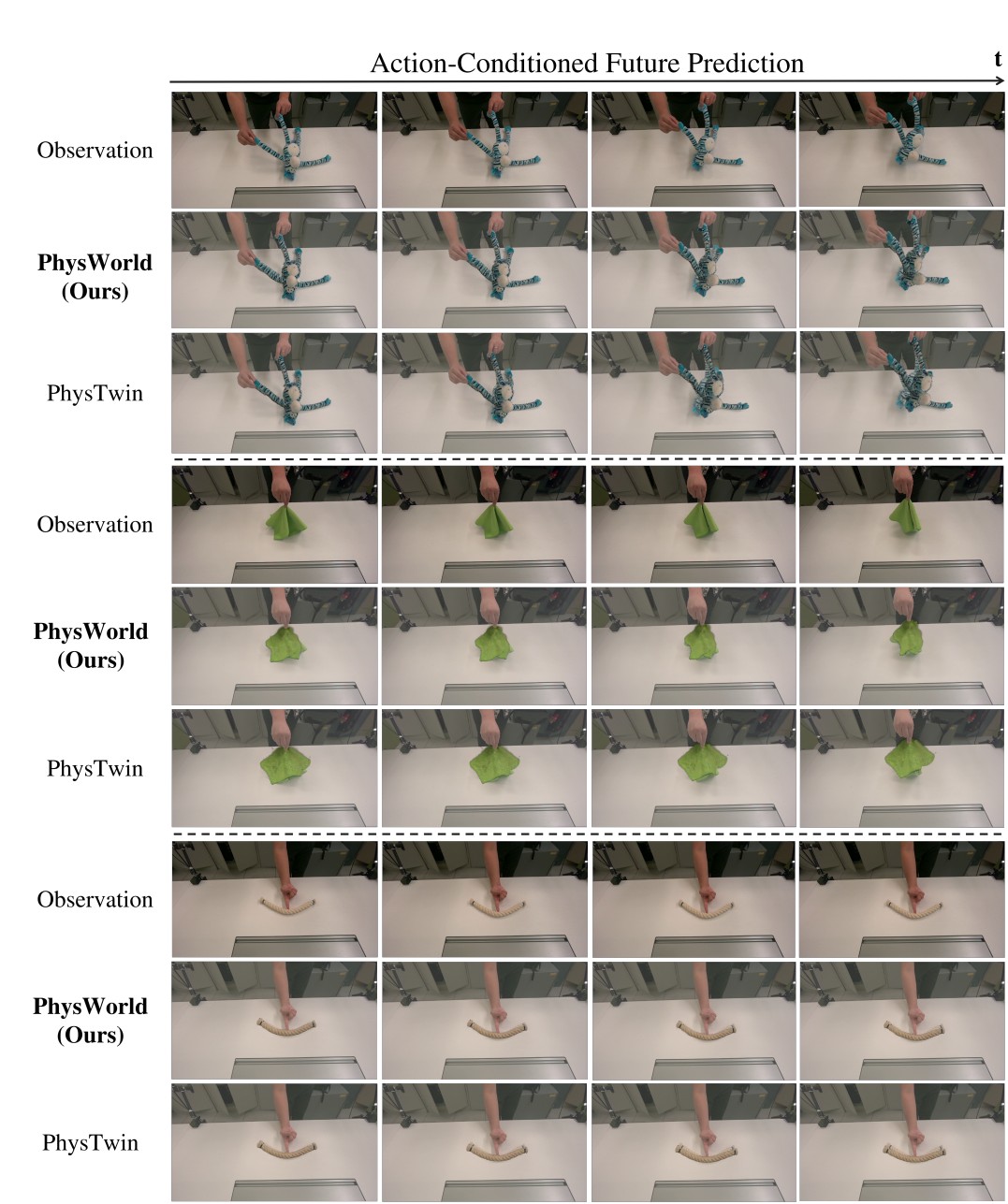

Figure 12: Additional visual results of action-conditioned future prediction. Our method's predicted positions show closer alignment with ground truth compared to PhysTwin.

