# OpenReview forum: "PhysWorld: From Real Videos to World Models of Deformable Objects via Physics-Aware Demonstration Synthesis"
_ICLR.cc/2026/Conference — Submitted to ICLR 2026_

### Official Review · Reviewer_oNhM · 2025-10-29

**Soundness:** 2
**Presentation:** 3
**Contribution:** 2
**Rating:** 4
**Confidence:** 5

**Summary:**

This paper focuses on reconstructing a digital twin of a deformable object from videos. Technically, this paper extends PhysTwin: (1) it refines physical motion simulation by a part-based physical property variation method and a tailored optimization method; (2) it improves simulation speed by distilling physics simulation onto a GNN for dynamics prediction. Experiments show that the proposed method is much faster in simulation, and the simulated motion improves over PhysTwin.

**Strengths:**

- Fast simulation that is useful for motion planning.
- Improved quality upon prior work.

**Weaknesses:**

- I'm not convinced by the proposed additional components including part-based perturbation and the optimization strategy. In particular, looking at the L343 and L344, by adding these somewhat complicated components, the benefits over PhysTwin are marginal. Do they really justify the complication added? To see this, I recommend a baseline: PhysTwin + the GNN distillation. If this baseline works equally well, then I think only the GNN distillation is needed.
- There are only 3 examples in video results. More video results are needed to be convincing.

**Questions:**

See above.

---

> ### Author Response · Authors · 2025-11-21
>
> Thank you for the valuable comments and suggestions! We appreciate your questions and hope our responses could address your concerns.
>
> ## Question 1: Benefit of VMP-Gen / $P^3$-Pert and Why Distill from MPM Rather Than PhysTwin
>
> Response:
>
> **First**, we would like to clarify what is being compared at the lines the reviewer refers to. The rows around L343–L344 show a direct comparison between PhysTwin and our calibrated MPM-based digital twin (“Our MPM”). At this stage, we only optimize physical parameters to obtain a physics-consistent digital twin in MPM. Our VMP-Gen and $P^3$-Pert modules are not used yet, since they are designed to generate diverse demonstrations from the calibrated simulator.
>
> **Second**, the benefit of VMP-Gen and $P^3$-Pert appears when we train the GNN on synthesized demonstrations. As shown by our ablations (see Tables 4 and 5), removing VMP-Gen and/or $P^3$-Pert and training only on replayed trajectories leads to a less accurate and less robust world model. In contrast, VMP-Gen samples diverse action/parameter combinations, and $P^3$-Pert introduces structured part-level property variations, together yielding richer and physically grounded training data, thus improving GNN dynamics.
>
> **Third**, following the reviewer’s suggestion, we added a “PhysTwin + GNN distillation” baseline on the *double_lift_sloth* scenario. Concretely, we: (1) take PhysTwin’s spring–mass simulator as the teacher; (2) generate training data with and without VMP-Gen&$P^3$-Pert; and (3) train the same GNN under exactly the PhysWorld protocol.
>
> The following results show two trends:
>
> 1. PhysTwin + GNN with VMP-Gen/$P^3$-Pert outperforms PhysTwin + GNN without them, confirming that our VMP-Gen and $P^3$-Pert modules are beneficial even when the teacher is PhysTwin.
> 2. Our MPM-based PhysWorld outperforms the PhysTwin-based distilled GNN (with VMP-Gen/$P^3$-Pert), indicating that MPM provides higher-quality demonstrations for distillation.
>
> | Method           | VMP-Gen & $P^3$-Pert | CD ↓   | Track ↓ | IoU ↑ | PSNR ↑ | SSIM ↑ | LPIPS ↓ |
> | ---------------- | ------------------ | ------ | ------- | ----- | ------ | ------ | ------- |
> | PhysTwin + GNN   | ×                  | 0.0296 | 0.0679  | 55.66 | 21.220 | 0.895  | 0.121   |
> | PhysTwin + GNN   | ✓                  | 0.0187 | 0.0320  | 67.87 | 22.686 | 0.911  | 0.082   |
> | MPM + GNN (Ours) | ✓                  | **0.0100** | **0.0154**  | **78.66** | **24.666** | **0.921**  | **0.067**   |
>
> We have incorporated the above analysis into our revised manuscript in **Appendix C.2**.
>
> **Finally**, we would like to emphasize why using MPM as the teacher is conceptually important. PhysTwin’s underlying dynamics model is a spring-mass system, which is primarily suited for elastic materials. In contrast, MPM can naturally handle a much wider range of behaviors, including elasto-plastic deformation, granular materials, and even fluids.
> In newly added experiments on elasto-plastic objects from RoboCraft[1], we find that after physics-parameter optimization, MPM produces significantly more realistic deformations than PhysTwin, especially under large and irreversible deformations. The quantitative results are shown below.
>
> | Method                        | Chamfer Loss ↓  | EMD Loss ↓   | Hausdorff Loss ↓   |
> | ----------------------------- | -----------: | ---------------: | ---------------------: |
> | PhysTwin |     0.0190 |         0.0320 |               0.1701 |
> | Our MPM       |     **0.0175** |   **0.0276** |       **0.0963** |
>
> The qualitative results are shown in Figure 10 in **Appendix D.6** in the revised version of our paper.
>
> Our goal is to distill a lightweight neural model from a powerful, general-purpose simulator; in this sense, MPM is a more capable and future-proof teacher. VMP-Gen and $P^3$-Pert make this distillation effective by exposing the GNN to diverse, physically meaningful variations in actions and spatially varying properties, rather than replaying fixed trajectories.
>
> Taken together, these results suggest that (1) VMP-Gen and $P^3$-Pert provide clear and reproducible benefits over a “PhysTwin + GNN” baseline, and (2) the MPM-based digital twin is a substantially more powerful teacher than the spring–mass model in PhysTwin.
>
> [1] Robocraft: Learning to see, simulate, and shape elasto-plastic objects in 3d with graph networks. RSS 2022.
>
> ## Question 2: Supplementary Demonstration Videos
>
> Response:
>
> More video demonstrations are now available on: https://anonymousiclr123.github.io/anonymous_physworld.github.io/ .

---

### Official Review · Reviewer_aLrE · 2025-10-30

**Soundness:** 3
**Presentation:** 2
**Contribution:** 2
**Rating:** 4
**Confidence:** 4

**Summary:**

The paper presents a framework for building physics-consistent world models of deformable objects directly from short real-world videos. It bridges the gap between high-fidelity physical simulation and efficient neural dynamics learning by combining a MPM-based digital twin with GNNs.

**Strengths:**

* The paper is clear and well-written
* The VLM-assisted constitutive model selection introduces a creative and automated way to align material modeling with observed deformation behaviors.
* Experiments on 22 scenarios demonstrate strong performance

**Weaknesses:**

* Heavy dependence on differentiable MPM simulation. However, the MPM-based simulator's sim-2-real gap is not well studied.
* The generalization tests, though promising, focus on limited or scenarios from one dataset; real-world complexities (e.g., lighting, occlusion) are underexplored.
* The impact and robustness of the VLM-based material selection are not deeply analyzed—especially in ambiguous or noisy video settings.

**Questions:**

* Recent paper (https://arxiv.org/abs/2506.23126) has shown that using transformer as dynamics model is better than GNNs. Could you include the results of comparison between these two structures?
* If a fully-optimized MPM model can serve as a digital twin of the real world?
* How does the system handle failure cases in the VLM-based constitutive model selection—does it fall back to default physics priors?

---

> ### Author Response · Authors · 2025-11-21
>
> Thank you for the valuable comments and suggestions! We appreciate your questions and hope our responses could address your concerns.
>
> ## Question 1: Sim-to-Real Gap of the Calibrated MPM Digital Twin
>
> Response:
>
> Here, we explicitly quantify the gap by following ways.
>
> (1) Quantitative evaluation of the calibrated MPM digital twin. In Table 1 of the main paper, we evaluate the calibrated MPM digital twin directly against real test frames using both 3D (Chamfer Distance, Track) and 2D (PSNR/SSIM/LPIPS/IoU) metrics. The row “Our MPM” corresponds to a optimized simulator driven by actions extracted from the real videos, and it achieves the best or second-best scores among all baselines. This indicates that, after applying our proposed global-to-local physical parameter identification scheme, the resulting MPM model serves as a physics-consistent digital twin with a relatively small sim-to-real error.
>
> (2) Validating the physical realism of synthesized demonstrations.
> To assess the physical realism of our synthesized demonstrations, we adopt the evaluation protocol of PhyGenBench[1]. This benchmark measures the consistency of object motion with 27 physical laws, including those related to gravity, friction, elastic forces, and other basic dynamics. We apply these metrics to both our generated demonstrations and the real ground-truth videos. Our synthesized demonstrations achieve an average Realism score of **1.6010**, which is very close to the real videos’ score of **1.6087**, indicating that the generated object motions are highly physically realistic. We have incorporated this analysis into our revised manuscript in **Appendix D.2**.
>
> Beyond these, we further fine-tune the GNN’s physical properties on real videos at the end of the pipeline, which narrows the sim-to-real gap(as shown in Table 1).
>
> [1] Towards world simulator: Crafting physical commonsense-based benchmark for video generation. ICML, 2025.
>
> ## Question 2: Robustness to Real-World Visual Complexities (Lighting & Occlusion)
>
> Response:
>
> Here, we clarify our model's handling and consideration of lighting and occlusion.
>
> * For lighting, PhysWorld predicts dynamics from 3D point clouds, not raw pixels. Lighting only affect the 3DGS appearance rendering, while our dynamics modules (MPM and GMM) sees only 3D particles and actions. Our point-cloud-based design makes the dynamics prediction inherently more robust to appearance changes.
>
> * For occlusion, multi-view reconstruction alleviates occlusion because missing regions in one view are observed in others. Our evaluation uses 22 real-world scenarios with diverse objects and interactions. These videos naturally include hand–object occlusions. Tables 1 and 2 report results averaged over all 22 scenes on seen and unseen interactions, respectively. It shows that PhysWorld generalizes better than PhysTwin to novel actions in real videos.

---

> > ### Author Response · Authors · 2025-11-21
> >
> > ## Question 3: Backbone Comparison: GNN vs. Transformer/MLP
> >
> > Response:
> >
> > We thank the reviewer for pointing us to this recent work. We have carefully read the paper of ParticleFormer[4]; however, since its implementation has not yet been open-sourced, reproducing the method is somewhat difficult.
> >
> > To address the reviewer’s concern, we implemented two alternative backbones by replacing our GNN with:
> >
> >  * An MLP-based architecture following PointMLP[2]
> >
> >  * A Transformer-based architecture following PointTransformerV3[3].
> >
> > We trained these variants on the *double\_lift\_sloth* scenario under the same training protocol and compared them against our GNN-based model. As shown in the following table, both the MLP-based and Transformer-based models underperform the GNN in terms of prediction accuracy, and their inference speed is also much slower. This supports our claim that a GNN backbone provides an efficient and accurate representation for modeling the dynamics of various deformable objects.
> >
> > | Methods        | CD↓        | Track↓     | IoU↑      | PSNR↑      | SSIM↑     | LPIPS↓    | FPS↑    |
> > | -------------- | ---------- | ---------- | --------- | ---------- | --------- | --------- | ------- |
> > | Transformer    | 0.0497     | 0.0656     | 42.43     | 20.291     | 0.891     | 0.133     | 56      |
> > | MLP            | 0.0213     | 0.0343     | 70.13     | 23.206     | 0.914     | 0.084     | 107     |
> > | GNN (Ours) | **0.0100** | **0.0154** | **78.66** | **24.666** | **0.921** | **0.067** | **784** |
> >
> > This may be because GNNs have a strong inductive bias for deformable dynamics: they model sparse, local particle–particle/particle–tool interactions via a radius graph and multi-step message passing, which PhysWorld also adopts for real-time, material-aware dynamics prediction.
> > Such graph structure aligns with how forces propagate in deformable media and generalizes better to changing topology/contacts, whereas this MLP must learn interactions implicitly and Transformers incur dense-attention overhead without explicit locality priors. Owing to these advantages, prior simulation-learning works have widely adopted GNN-based dynamics models for diverse deformable materials such as plasticine, cloth, and fluids, and we follow this established and effective design. We have incorporated the above analysis into our revised manuscript in **Appendix C.1**.
> >
> > Furthermore, we also believe that with careful and targeted design, the Transformer-based architecture may have the potential to outperform GNNs. We will conduct a comparison as soon as ParticleFormer[4] is open-sourced.
> >
> > [2] Rethinking network design and local geometry in point cloud: A simple residual MLP framework. ICLR, 2022.
> >
> > [3] Point Transformer v3: Simpler faster stronger. CVPR, 2024.
> >
> > [4] ParticleFormer: A 3D Point Cloud World Model for Multi-Object, Multi-Material Robotic Manipulation. CoRL, 2025.
> >
> > ## Question 4: Can a Fully Optimized MPM Alone Serve as a Practical Digital Twin?
> >
> > Response:
> >
> > In principle, a fully-optimized MPM model with the correct constitutive law and parameters can be a very strong digital-twin–style reference. However, its main limitation is computational cost, especially when being used for model-based planning that requires a large number of rollouts.
> >
> > To make this concrete, we directly compared using the MPM-based digital twin and our GNN-based PhysWorld as the dynamics model inside an MPPI planner, tasked with moving the object from an initial configuration to a target configuration. We use a standard MPPI setup on a single RTX 4060 Ti (16 GB) GPU, with 2,000 randomly sampled trajectories per planning step and 10 planning steps in total. Under this setting: the MPM-based digital twin requires on average **4,687s** per planning step, which makes it effectively unusable for interactive planning; in contrast, our GNN-based PhysWorld needs only **22s** per planning step. **We have provided more planning-related details in our response (Question 1) to Reviewer YYty, and kindly refer to it for details**
> >
> > This experiment shows that, even if the MPM simulator is well calibrated, using it directly as the world model in a planner is prohibitively slow, whereas our GNN-based world model makes the same planning problem more efficient.

---

> > > ### Author Response · Authors · 2025-11-21
> > >
> > > ## Question 5: Robustness of VLM-Assisted Constitutive Model Selection
> > >
> > > Response:
> > >
> > > To further evaluate the robustness of our VLM-based constitutive model selection, we collected additional videos from the Internet showing hand/robot-arm interactions with deformable objects (as shown in Figure 7) and asked Qwen3 to choose the most suitable constitutive model for the manipulated object from our material library, containing 4 elastic models and 4 plastic models. We then compared Qwen3’s predictions with expert annotations and observed consistently high agreement (100\%) accuracy on this extended set. This suggests that, with a reasonably designed (and not overly large) material library, constitutive-model selection using a modern VLM is a manageable subproblem in our pipeline.
> > >
> > > We also note that picking a “second-best” model does not necessarily break the system. In practice, some objects can be described reasonably well by more than one constitutive law. For instance, many cloth-like objects are best matched by an anisotropic hyperelastic model, but a simpler Neo-Hookean elastic model can still yield plausible behavior for thicker fabrics. To test robustness, we conducted this study on the *single\_lift\_cloth_1* scene, where the theoretically optimal constitutive model is Anisotropic Hyperelastic. We intentionally replaced it with a Neo-Hookean model and reran the pipeline. The results below show that our framework can re-optimize the physical parameters under this suboptimal choice and still recover accurate dynamics, indicating tolerance to occasional non-optimal model selection. We have incorporated the above analysis into our revised manuscript in **Appendix D.3**.
> > >
> > > | Methods                     | CD↓        | Track↓     | IoU↑      | PSNR↑      | SSIM↑     | LPIPS↓    |
> > > | --------------------------- | ---------- | ---------- | --------- | ---------- | --------- | --------- |
> > > | Ours (Optimal material) | **0.0186** | **0.0241** | **75.54** | **27.700** | **0.974** | **0.039** |
> > > | Ours (Suboptimal material)  | 0.0193     | 0.0339     | 62.15     | 26.512     | 0.968     | 0.046     |
> > > | PhysTwin                    | 0.0221     | 0.0286     | 55.75     | 23.919     | 0.958     | 0.062     |

---

### Official Review · Reviewer_WAMx · 2025-10-31

**Soundness:** 3
**Presentation:** 3
**Contribution:** 2
**Rating:** 4
**Confidence:** 4

**Summary:**

The paper proposes to learn world models of deformable objects by grounding videos rendered from a physical simulator. The authors builds material-point methods simulator to generate data with physical plausibility, in contrast to learning from limited real world video streams. The demonstrations collected from the simulator are then used to train a graph neural network that amortizes the prediction, with the physical properties as the embedded attributes. With an inverse rendering pipeline, real videos are used to fine-tune the physical properties for a better alignment with the reality. The results show faster prediction due to the graph neural network inference and better accuracy comparing a baseline method.

**Strengths:**

* Well articulated approaches and easy to grasp the central idea;
* Modelling and predicting deformable object behavior is an important topic so the paper may bring about impacts;
* Promising to publish code and pre-trained models. Important for reproducibility of the work.

**Weaknesses:**

* All technical methods are well known and the paper reads like a concatenation of them without much scientific novelty or insights;
* Validation should be stronger. Especially to the point of using a GNN to amortize a simulation. Better to have more direct evidence to show the benefits of having a faster predictive model when the simulator is already there. See more elaborated points in questions.
* Validation should also be more extensive. I think this is especially the case given the paper is more like a system paper with slim scientific contributions. The tested scenes look a bit trivial and hardly representative of interesting deformable tasks.

**Questions:**

* What is the scientific observation or insight beyond the system work presented in the paper?
* What is the point of using a neural network to fit a simulator when the latter is already readily available for generating as much data as needed? If the argument lies in prediction efficiency, can this efficiency be shown necessary in some applications, such as real-time control or speeding-up reinforcement learning environment queries?
* Can the results significance be demonstrated with a statistical analysis as many numbers in tables look very close to each other?
* The performance is most demonstrated as the prediction quality while it is hard to tell whether this is necessarily translating into a good model for policy learning or model-based planning. Can experiments like in Figure 4 be expanded to more realistic and extensive scenarios?

---

> ### Author Response · Authors · 2025-11-21
>
> Thank you for the valuable comments and suggestions! We appreciate your questions and hope our responses could address your concerns.
>
> ## Question 1: Key Scientific Observations and Insights
>
> Response:
>
> Beyond the concrete system design, our main scientific insight is about how real videos, physics simulators, and efficient neural world models should interact when only short interaction sequences of real deformable objects are available.
>
> Concretely, we observe that:
>
> * Short real videos are too weak to directly learn efficient neural world models, but are strong enough to identify physical parameters. When we train a neural dynamics model purely on the limited real trajectories, it overfits individual scenes and fails to generalize to new interactions (see Table 6). In contrast, using the same videos to calibrate a physically-based simulator (by estimating spatially varying stiffness, damping, friction, etc.) and then using this calibrated simulator as a data factory produces a world model that generalizes much better across unseen interactions and objects.
>
> * Physical plausibility and diversity of the synthesized data are both crucial to learn satisfactory world models. Naïve synthetic data (e.g., random trajectories or global parameter jitter) does not suffice: the learned model still misses many realistic behaviors. The physically informed components we introduce (such as curvature- and velocity-aware trajectory generation as well as part-aware, spatially correlated perturbations of material parameters) in the calibrated simulator lead to significantly more diverse yet still physically plausible demonstrations, and this directly translates into higher prediction fidelity (see Tables 4 and 5).
>
> * Once calibrated, the simulator is best used as a generator, not as an online predictor. After we have identified the physical properties from real videos, the role of the simulator changes. It becomes a one-time “teacher” for generating rich synthetic rollouts, while the neural world model (i.e., GNN) becomes the “student” that is fast enough to be used inside real-time planners or model-based control.
>
> ## Question 2: Why a GNN-Based World Model Is Needed Beyond Calibrated MPM
>
> Response:
>
> Our goal is not to replace the MPM simulator, but to change its role: we use the simulator offline to calibrate physical parameters from short real videos and to generate diverse, physically consistent demonstrations, and then learn an amortized neural world model that is fast enough to be used inside planners that require thousands of rollouts.
>
> Second, we explicitly demonstrated that this efficiency was necessary in a concrete model-based planning application. We used a standard MPPI planner and evaluated the relocating task on a single RTX 4060 Ti (16 GB) GPU, with 2,000 randomly sampled trajectories per planning step and 10 planning steps in total. Under this setting:
>
> * the MPM-based digital twin required on average 4,687s per planning step,
> * PhysTwin still needed 998s per step,
>
> making both methods effectively unusable for interactive planning. In contrast, our GNN-based PhysWorld needed only 22s per planning step, thanks to its much higher inference speed and the ability to evaluate many trajectories in parallel. **We have also provided more planning-related details in our response (Question 1) to Reviewer YYty, and kinldly refer to it for details**.
> This experiment shows that a faster neural world model is not just a cosmetic improvement over the simulator, but a practical requirement for embedding deformable-object dynamics into real-time (or near–real-time) planners.

---

> > ### Author Response · Authors · 2025-11-21
> >
> > ## Question 3: Interpreting the Significance of the Quantitative Metrics
> >
> > Response:
> >
> > This work aims to improve inference speed while maintaining and even improving future prediction performance.
> > As shown in Table 1, our PhysWorld offers a substantial inference speed advantage over both MPM and PhysTwin—approximately 400× faster than MPM and 47× faster than PhysTwin—which makes it much more suitable as a practical world model for downstream tasks such as model-based planning.
> >
> > ## Question 4: Evidence for Downstream Model-Based Planning Benefits
> >
> > Response:
> >
> > In the previous submission, Figure 4 in our paper already demonstrates that PhysWorld can be integrated into an MPPI planner to manipulate a rope from an initial configuration to a specified target configuration. In the revised version, we have provided additional planning examples and report quantitative planning metrics, including: (1) curves of the distance error between the object’s current and target configurations with respect to the planning steps, (2) success-rate curves under different error thresholds, and (3) a comparison of the wall-clock time required for 10-step MPPI planning in **Appendix D.1**.
> > The results show that, thanks to the GNN’s ability to perform parallel inference over multiple trajectories, our method requires significantly less planning time while maintaining satisfactory control performance.

---

> > > ### Comment · Reviewer_WAMx · 2025-11-26
> > >
> > > Thank the authors for the detailed rebuttal. I think the stated insights make sense while also not very surprising. Essentially, it sounds like "fitting a neural model on a small dataset can benefit from augmented data generated from another model, with stronger priors/structures and calibration on the small data". This might be useful from specific application perspective but limited in terms of novelty in general machine learning. In light of this, I believe the paper can be best sold as some enabling technique with strong application prospects, for which the paper's validation, including the appended one, looks slim. The paper could benefit much from tasks with clearer contexts and essentials beyond reconfiguring simple deformables.

---

> > > > ### Author Response · Authors · 2025-11-27
> > > >
> > > > We appreciate the reviewer's prompt responses and recognition of our rebuttal. We would like to clarify some concerns further here.
> > > >
> > > > 1. **Question:** Essentially, it sounds like "fitting a neural model on a small dataset can benefit from augmented data generated from another model, with stronger priors/structures and calibration on the small data".
> > > >
> > > >    **Response:**
> > > >
> > > >
> > > >    We responded using some easy-to-understand descriptions in the previous rebuttal, but this may have led to some misunderstandings. Here we further clarify the innovative aspects.
> > > >
> > > >    **First**, we emphasize that the critical determinant for a world model trained from small dataset is the **physical consistency** of the object's properties (e.g., material and Young's modulus) and motion laws with the real world in the synthesized demonstrations. To ensure the constructed MPM-based digital twin is physics-consistent, we propose the **VLM-assisted constitutive model selection** and **Global-to-Local optimization** approaches. This goes beyond vanilla augmentation by identifying heterogeneous physical properties from sparse observations to bridge the sim-to-real gap (see Table 3).
> > > >
> > > >    **Second**, we emphasize that the **diversity** of demonstrations is also crucial, thus we introduce **VMP-Gen** and **$P^3$-Pert**.
> > > >    They adds trajectory, velocity, and part-aware property augments to the digital twin in the MPM, improving the diversity of demonstrations while maintaining physical consistency as much as possible (see Tables 4 and 5).
> > > >
> > > >    As far as we know, this perspective of **physically consistent yet diverse synthesis** hasn't been considered in previous works.
> > > >
> > > > 2. **Question:** This might be useful from specific application perspective but limited in terms of novelty in general machine learning.
> > > >
> > > >    **Response:**
> > > >
> > > >    Thank you for recognizing its practicality. We argue that solving high-dimensional dynamics from short-term data is a critical and non-trivial problem in robot learning. PhysWorld operates on just **1-10 seconds of video**. It avoids the high cost of hardware wear-and-tear or labor-intensive data collection. We are also eager to explore its applications in other fields in the future.
> > > >
> > > > 3. **Question:** I believe the paper can be best sold as some enabling technique with strong application prospects, for which the paper's validation, including the appended one, looks slim. The paper could benefit much from tasks with clearer contexts and essentials beyond reconfiguring simple deformables.
> > > >
> > > >    **Response:**
> > > >
> > > >    We adopt the standard benchmarks in this field, where baselines like GS-Dynamics (CoRL 2024), AdaptiGraph (RSS 2024), and PhysTwin (ICCV 2025) all utilize object reconfiguration to validate prediction accuracy. In physics simulation, seemingly "simple" objects like ropes and cloth actually involve infinite degrees of freedom, self-collisions, and complex dynamics. Solving this problem efficiently and accurately is not easy. We also extend validation to **elasto-plastic objects** (see Figure 11 and Table 12), involving irreversible deformations that spring-mass models like PhysTwin fail to capture.
> > > >
> > > >    Furthermore, in the comparison to PhysTwin, our PhysWorld performs **47x faster**. Figures 4, 5, 6 and Table 9 show that the improvements of PhysWorld can indeed translate to faster and more accurate real-world control, which stands as the ultimate test for world models.

---

### Official Review · Reviewer_kicv · 2025-11-01

**Soundness:** 2
**Presentation:** 2
**Contribution:** 2
**Rating:** 4
**Confidence:** 3

**Summary:**

This paper proposes PhysWorld, a framework for learning accurate and efficient world models of deformable objects from short, real-world videos. It tries to address the trade-off between high-fidelity, physics-based simulators (which are accurate but slow) and learning-based models (which are fast but data-hungry and often physically inconsistent). The proposed framework includes two augmentation methods Various Motion Pattern Generation (VMP-Gen) and Part-aware Physical Property Perturbation (P3-Pert) for increasing the diversity of synthesized demonstrations.

**Strengths:**

The paper addresses a practical problem: learning usable dynamic models from limited, real-world data. The proposed method offers a compelling solution that addresses the issues inherent in existing methods: the accuracy of a high-fidelity physics engine and the speed of a lightweight neural network. The demonstrated 47x speedup over the SOTA baseline is a massive practical gain that unlocks real-time applications, as evidenced by the successful MPPI planning experiment.

P3-Pert (Part-aware Physical Property Perturbation) utilizes semantic part features to generate spatially correlated noise for physical properties, which is far more physically plausible than naive random perturbations. It appears to be an effective method for generating meaningful data diversity, which the ablations confirm is superior. Using a VLM to automate the selection of a physics model from a library based on video is a valuable idea that simplifies a traditionally manual and expertise-driven process. Lastly, the two-stage optimization strategy is a methodologically sound and robust approach to parameter estimation, avoiding the pitfalls of optimizing all parameters from a cold start.

**Weaknesses:**

1) Lacking robustness analysis for 3D data preprocessing:

This paper proposes a pipeline for constructing world models from video. The entire pipeline begins with extraction of object point clouds from real interaction videos. This is a critical, non-trivial preprocessing step. The quality (density, noise, completeness) of this initial 3D tracking is fundamental to the accuracy of the digital twin optimization. The paper does not analyze the framework's sensitivity to this input quality. If the tracking is poor, it's likely the digital twin will be inaccurate, and this error will propagate through the entire system.

2) Lacking analysis for external models:

The framework's novelty relies on two other sophisticated models: a VLM (Qwen3) for model selection and a PartField model. This introduces potential failure points. What happens if the VLM misidentifies the material? Or if the PartField model provides a poor or trivial segmentation for a new object category? The paper doesn't discuss the robustness of the pipeline to failures in these helper modules.

**Questions:**

1) Robustness to VLM Failure:

The 100% accuracy for the VLM model selection is noted in Appendix B2. What will happen if a suboptimal model is chosen? For instance, if the VLM mistakenly selects "Neo-Hookean" for the "cloth" object, how much does this error degrade the final accuracy of the fine-tuned PhysWorld GNN?

2) Setup cost:

The paper introduces the fast inference speed of the proposed method. However, the training/preparation cost would be important for practitioners. Could you please provide an approximate wall-clock time for the entire setup process for a new object? (e.g., for the "double lift sloth" scene). This would include digital twin optimization, demonstration synthesis, and GNN training/fine-tuning. This would provide a complete picture of the method's computational cost.

---

> ### Author Response · Authors · 2025-11-21
>
> Thank you for the valuable comments and suggestions! We appreciate your questions and hope our responses could address your concerns.
>
> ## Question 1: Sensitivity to 3D Tracking and Point-Cloud Quality
>
> Response:
>
> We agree that the initial extraction and tracking of object point clouds from multi-view RGB-D videos is a critical preprocessing step. In our implementation, this module follows the pipeline of PhysTwin, which has already been carefully engineered and extensively validated on various real-world interaction scenarios. Concretely, PhysTwin first turns multi-view RGB-D videos into partial 3D point clouds. To deal with sparse, noisy, and occluded observations, it uses a generative 3D shape prior together with a coarse-to-fine registration process. It then refines the 3D motion trajectories by lifting 2D tracks from CoTracker3 into 3D, and jointly optimizing the object’s geometry and dynamics using physics-based and rendering-based losses. These components are shown in PhysTwin to produce stable reconstructions and accurate resimulation/future prediction even under limited viewpoints and partial occlusions.
>
> In our work, for a fair comparison, we reuse this perception and tracking pipeline following PhysTwin. We mainly focus our contributions on the subsequent stages: VLM-assisted consitutive model selection and MPM physical property optimization, physics-aware demonstration synthesis, and efficient deformable world-model learning. In practice, we do observe that some tracked points can exhibit jitter or local inaccuracies. To mitigate this, when using tracking signals to supervise MPM parameter optimization and to fine-tune the GNN, we only rely on stable, high-confidence trajectories: intuitively, a track is treated as reliable if its motion is consistent with that of its spatial neighbors over time. Furthermore, even when individual correspondences are imperfect, our loss for physical parameter optimization and GNN fine-tuning includes a Chamfer distance term between simulated and observed point clouds, which measures set-level similarity without requiring precise one-to-one matches between points. This design makes the optimization more robust to local tracking noise and occasional mismatches.
>
> ## Question 2: Robustness of VLM-Assisted Constitutive Model Selection
>
> Response:
>
> To further evaluate the robustness of our VLM-based constitutive model selection, we collected additional videos from the Internet showing hand/robot-arm interactions with deformable objects (as shown in Figure 7) and asked Qwen3 to choose the most suitable constitutive model for the manipulated object from our material library, containing 4 elastic models and 4 plastic models. We then compared Qwen3’s predictions with expert annotations and observed consistently high agreement (100\%) accuracy on this extended set. This suggests that, with a reasonably designed (and not overly large) material library, constitutive-model selection using a modern VLM is a manageable subproblem in our pipeline.
>
> We also note that picking a “second-best” model does not necessarily break the system. In practice, some objects can be described reasonably well by more than one constitutive law. For instance, many cloth-like objects are best matched by an anisotropic hyperelastic model, but a simpler Neo-Hookean elastic model can still yield plausible behavior for thicker fabrics. To test robustness, we conducted this study on the *single\_lift\_cloth_1* scene, where the theoretically optimal constitutive model is Anisotropic Hyperelastic. We intentionally replaced it with a Neo-Hookean model and reran the pipeline. The results below show that our framework can re-optimize the physical parameters under this suboptimal choice and still recover accurate dynamics, indicating tolerance to occasional non-optimal model selection. We have incorporated the above analysis into our revised manuscript in **Appendix D.3**.
>
> | Methods                     | CD↓        | Track↓     | IoU↑      | PSNR↑      | SSIM↑     | LPIPS↓    |
> | --------------------------- | ---------- | ---------- | --------- | ---------- | --------- | --------- |
> | Ours (Optimal material) | **0.0186** | **0.0241** | **75.54** | **27.700** | **0.974** | **0.039** |
> | Ours (Suboptimal material)  | 0.0193     | 0.0339     | 62.15     | 26.512     | 0.968     | 0.046     |
> | PhysTwin                    | 0.0221     | 0.0286     | 55.75     | 23.919     | 0.958     | 0.062     |

---

> > ### Author Response · Authors · 2025-11-21
> >
> > ## Question 3: Robustness to PartField Feature Quality
> >
> > Response:
> >
> > Regarding the PartField module, our pipeline does not rely on a perfectly discrete, semantically correct part segmentation. We only use PartField’s continuous intermediate features to induce spatially coherent perturbations: points with similar features receive similar physical-parameter perturbations. This yields smooth, non-uniform material fields without sharp discontinuities, so the MPM simulator remains stable.
> >
> > Even if the PartField features are not fully accurate, they still provide useful non-uniform physical-property examples for demonstration synthesis, which is exactly what we need to train the GNN to cope with heterogeneous materials. Moreover, the mismatch is further mitigated because we fine-tune the object’s physical parameters on real videos later, so the final calibrated properties are guided by real dynamics. Empirically, Table 5 also supports this design: using PartField as the part-feature extractor for $P^3$-Pert leads to a clear improvement in the trained GNN’s prediction accuracy, demonstrating its practical benefit.
> >
> > In the worst case where the features became nearly trivial for a new category, the perturbations would collapse to being almost uniform—still safe and stable, just with reduced motion diversity. Importantly, we did not observe such a degenerate-feature case in any of our tested scenarios.
> >
> > ## Question 4: Overall Computational Cost
> >
> > Response:
> >
> > We provide a new table that reports the overall computational cost for a representative scene, *double\_lift\_sloth*, on a single RTX 4060 Ti (16 GB) GPU. Concretely, the pipeline takes on average
> > (1) **0.38 h (4.3%)** for the MPM-based digital twin optimization,
> > (2) **3.00 h (33.7%)** for demonstration synthesis,
> > (3) **5.48 h (61.7%)** for GNN training, and
> > (4) **0.03 h (0.3%)** for GNN fine-tuning on real videos,
> > for a total of **8.89 h (~8.9 h)** per object.
> >
> > | Stage                   | Duration (h) | Share of total |
> > | ----------------------- | ------------ | -------------- |
> > | MPM optimization        | 0.38         | 4.3%           |
> > | Demonstration synthesis | 3.00         | 33.7%          |
> > | GNN training            | 5.48         | 61.7%          |
> > | GNN fine-tuning         | 0.03         | 0.3%           |
> > | **Total**               | **8.89**     | **100%**       |
> >
> > While this one-time preparation cost is non-trivial, it yields a fast world model that runs orders of magnitude faster than the underlying MPM simulator. The fast world model can be directly used for downstream tasks such as model-based planning and control, where high inference speed is crucial. We have incorporated the planning and control results into our revised manuscript in **Appendix D.5**. **We have also provided more details in our response (Question 1) to Reviewer YYty, and kindly refer to it for details**.

---

### Official Review · Reviewer_YYty · 2025-11-01

**Soundness:** 3
**Presentation:** 3
**Contribution:** 3
**Rating:** 6
**Confidence:** 3

**Summary:**

This paper presents PhysWorld, a framework for constructing physics-consistent and efficient world models of deformable objects from short real-world videos.
The method first builds a digital twin in an MPM simulator, where a Vision-Language Model (Qwen3) selects the appropriate constitutive material models and a global-to-local optimization refines physical parameters (e.g., friction, density, Young’s modulus).
The calibrated simulator then generates diverse synthetic demonstrations via Various Motion Pattern Generation (VMP-Gen) and Part-aware Physical Property Perturbation (P³-Pert).
A GNN-based world model is trained on these demonstrations and fine-tuned using real videos.
Experiments show that PhysWorld achieves accurate predictions and 47× faster inference than PhysTwin while maintaining visual and physical consistency.
The paper also provides a qualitative demonstration of model-based planning using MPPI for rope manipulation.

**Strengths:**

1. **Comprehensive and well-integrated framework**
   The paper thoughtfully combines VLM-based material selection, multi-stage parameter optimization, and physics-guided data augmentation into a coherent pipeline bridging simulation and learning.

2. **Strong accuracy–efficiency trade-off**
   The proposed GNN-based world model achieves high predictive accuracy while running at **799 FPS**, demonstrating its potential for real-time applications that require fast yet physically consistent inference.

3. **Visual prediction capability**
   Integration of 3D Gaussian Splatting and Linear Blend Skinning enables **action-conditioned video generation**, evaluated with PSNR, SSIM, and LPIPS metrics.

4. **Generalization to unseen interactions**
   The model generalizes well to new manipulation sequences and shows physically plausible deformation behavior (Fig. 3, Table 2).

5. **Detailed ablation studies**
   The effects of global-to-local optimization, VMP-Gen, and P³-Pert are clearly quantified (Tables 3–5), and the results support the design choices.

**Weaknesses:**

1. **Lack of quantitative downstream (control) evaluation — a key remaining limitation**
   While the framework’s real-time capability is convincingly demonstrated (47× faster inference), the paper does not provide **quantitative results in downstream control or planning tasks**.
   The MPPI example (Fig. 4) is qualitative only. Demonstrating success rates, trajectory errors, or computation times in model-based control would substantially strengthen the practical significance.
   Given that real-time inference is the method’s major advantage, connecting it to tangible control improvements would enhance the paper’s overall impact.

2. **Action-conditioned requirement**
   The world model relies on explicit control inputs (\(a_t\)) extracted from video, and learning from action-free observational data is not addressed. This limits applicability to passive video datasets.

3. **Architectural diversity**
   The study focuses solely on GNNs. Comparisons with alternative architectures (e.g., Transformers or MLP-based dynamics models) could clarify whether the observed advantages are model-specific or framework-driven.

4. **Realism of synthetic demonstrations**
   Although the proposed perturbation and trajectory generation methods improve diversity, there is no quantitative analysis of how closely these synthetic motions resemble real physical dynamics.

5. **VLM-based model selection robustness**
   The Qwen3-based constitutive model selection is promising, but evaluation is limited to 22 clean scenarios. Its reliability under noise, occlusion, or mixed-material settings remains unclear.

6. **Scalability and physical consistency metrics**
   Experiments are conducted on relatively small particle systems (≈100–150 nodes). Performance and stability on larger, multi-object, or more complex scenes are not analyzed.
   Additionally, metrics directly assessing physical-law consistency (e.g., conservation of momentum or energy) are not reported.

7. **Reproducibility details**
   While optimization procedures are well described, training sensitivity (e.g., to initialization, hyperparameters) and convergence analyses are not provided, making it difficult to assess robustness.

8. **Conceptual scope of “World Model”**
   The method models object-level deformable dynamics rather than a full scene-level world model. Clarifying this scope would help set appropriate expectations.

**Questions:**

1. Could you provide **quantitative control results** (e.g., success rate, trajectory error, or control frequency) to demonstrate how PhysWorld’s high inference speed translates to better control performance?
2. How physically realistic are the synthetic demonstrations generated by VMP-Gen and P³-Pert?
3. How stable is the **VLM-based material selection** under noisy or complex real-world videos?
4. How transferable are the fine-tuned physical parameters (Φ) to new objects or scenes?
5. How does inference speed scale with larger particle counts or multiple interacting objects?
6. Could the model be extended to **action-free settings** through action inference or inverse dynamics?
7. Have you considered comparing the GNN to a Transformer-based or MLP-based world model for efficiency and accuracy?
8. Would adding explicit physical consistency losses (e.g., for momentum or volume preservation) further improve realism?

---

> ### Author Response · Authors · 2025-11-21
>
> Thank you for the valuable comments and suggestions! We appreciate your questions and hope our responses could address your concerns.
>
> ## Question 1: Quantitative MPPI Planning Evaluation
>
> Response:
>
> We use an MPPI-based planner as follows. At each planning step, we sample 2000 candidate action trajectories and roll them out with our learned dynamics model. We then measure how far each trajectory’s final state is from the target, give higher weights to trajectories with smaller final errors, and compute the control as a weighted average of all trajectories. We execute only the first action of this averaged sequence, then re-sample and repeat the process in a receding-horizon manner.
>
> Based on this planning setup, we have reported quantitative metrics including (1) curves of the distance error between the object’s current and target positions with respect to the planning steps, (2) success-rate curves with respect to several discrete error thresholds, and (3) a comparison of the wall-clock time required for 10-step MPPI planning, where each step samples 2000 trajectories. Thanks to the GNN’s ability to perform parallel inference over multiple trajectories, our method requires significantly less planning time while still achieving strong control performance. These results (Figure 5, Figure 6, Table 9) have been added to **Appendix D.1** in the revised version of the paper.
>
> The following results compare the planning time of our method against PhysTwin and MPM.
>
> | Methods | PhysTwin | MPM | GNN (Ours) |
> | --- | --- | --- | --- |
> | Planning time per step (s) | 998 | 4687 | **22** |
>
> ## Question 2: Applicability to Action-Free Dynamics
>
> Response:
>
> Our primary goal is to enable controllable manipulation of deformable objects. Therefore, we position PhysWorld as an action-conditioned world model and align our experimental setup with prior action-conditioned baselines such as PhysTwin. Extending PhysWorld to purely action-free observational data is a natural and interesting extension, and we will highlight it as a promising direction for future work.
>
> ## Question 3: Backbone Ablation: GNN vs. Transformer/MLP
>
> Response:
>
> To address the reviewer’s concern, we have implemented two alternative backbones by replacing our GNN with:
>  * An MLP-based architecture following PointMLP[1]
>
>  * A Transformer-based architecture following PointTransformerV3[2].
>
> We trained these variants on the *double\_lift\_sloth* scenario under the same training protocol and compared them against our GNN-based model. As shown in the following table, both the MLP-based and Transformer-based models underperform the GNN in terms of prediction accuracy, and their inference speed is also much slower. This supports our claim that a GNN backbone provides an efficient and accurate representation for modeling the dynamics of various deformable objects.
>
> | Methods        | CD↓        | Track↓     | IoU↑      | PSNR↑      | SSIM↑     | LPIPS↓    | FPS↑    |
> | -------------- | ---------- | ---------- | --------- | ---------- | --------- | --------- | ------- |
> | Transformer    | 0.0497     | 0.0656     | 42.43     | 20.291     | 0.891     | 0.133     | 56      |
> | MLP            | 0.0213     | 0.0343     | 70.13     | 23.206     | 0.914     | 0.084     | 107     |
> | GNN (Ours) | **0.0100** | **0.0154** | **78.66** | **24.666** | **0.921** | **0.067** | **784** |
>
> This may be because GNNs have a strong inductive bias for deformable dynamics: they model sparse, local particle–particle/particle–tool interactions via a radius graph and multi-step message passing, which PhysWorld also adopts for real-time, material-aware dynamics prediction.
> Such graph structure aligns with how forces propagate in deformable media and generalizes better to changing topology/contacts, whereas MLPs must learn interactions implicitly and Transformers incur dense-attention overhead without explicit locality priors. Owing to these advantages, prior simulation-learning works have widely adopted GNN-based dynamics models for diverse deformable materials such as plasticine, cloth, and fluids, and we follow this established and effective design. We have incorporated the above analysis into our revised manuscript in **Appendix C.1**.
>
> [1] Rethinking network design and local geometry in point cloud: A simple residual MLP framework. ICLR 2022.
>
> [2] Point transformer v3: Simpler faster stronger. CVPR 2024.

---

> > ### Author Response · Authors · 2025-11-21
> >
> > ## Question 4: Physical Realism of Synthesized Demonstrations
> >
> > Response:
> >
> > To assess the physical realism of our synthesized demonstrations, we adopt the evaluation protocol of PhyGenBench[3]. This benchmark measures the consistency of object motion with 27 physical laws, including those related to gravity, friction, elastic forces, and other basic dynamics. We have applied these metrics to both our generated demonstrations and the real ground-truth videos. Our synthesized demonstrations achieved an average Realism score of **1.6010**, which is very close to the real videos’ score of **1.6087**, indicating that the generated object motions are highly physically realistic. We have incorporated the above analysis into our revised manuscript in **Appendix D.2**.
> >
> > [3] Towards world simulator: Crafting physical commonsense-based benchmark for video generation. ICML 2025.
> >
> >
> > ## Question 5: Robustness of VLM-Assisted Constitutive Model Selection
> >
> > Response:
> >
> > To further evaluate the robustness of our VLM-based constitutive model selection, we collected additional videos from the Internet showing hand/robot-arm interactions with deformable objects (as shown in Figure 7) and asked Qwen3 to choose the most suitable constitutive model for the manipulated object from our material library, containing 4 elastic models and 4 plastic models. We then compared Qwen3’s predictions with expert annotations and observed consistently high agreement (100\%) accuracy on this extended set. This suggests that, with a reasonably designed (and not overly large) material library, constitutive-model selection using a modern VLM is a manageable subproblem in our pipeline.
> >
> > We also note that picking a “second-best” model does not necessarily break the system. In practice, some objects can be described reasonably well by more than one constitutive law. For instance, many cloth-like objects are best matched by an anisotropic hyperelastic model, but a simpler Neo-Hookean elastic model can still yield plausible behavior for thicker fabrics. To test robustness, we conducted this study on the *single\_lift\_cloth_1* scene, where the theoretically optimal constitutive model is Anisotropic Hyperelastic. We intentionally replaced it with a Neo-Hookean model and reran the pipeline. The results below show that our framework can re-optimize the physical parameters under this suboptimal choice and still recover accurate dynamics, indicating tolerance to occasional non-optimal model selection. We have incorporated the above analysis into our revised manuscript in **Appendix D.3**.
> >
> > | Methods                     | CD↓        | Track↓     | IoU↑      | PSNR↑      | SSIM↑     | LPIPS↓    |
> > | --------------------------- | ---------- | ---------- | --------- | ---------- | --------- | --------- |
> > | Ours (Optimal material) | **0.0186** | **0.0241** | **75.54** | **27.700** | **0.974** | **0.039** |
> > | Ours (Suboptimal material)  | 0.0193     | 0.0339     | 62.15     | 26.512     | 0.968     | 0.046     |
> > | PhysTwin                    | 0.0221     | 0.0286     | 55.75     | 23.919     | 0.958     | 0.062     |

---

> > > ### Author Response · Authors · 2025-11-21
> > >
> > > ## Question 6: Scalability and Physical-Law Consistency
> > >
> > > Response:
> > >
> > > In our real-world setups, we empirically found that using a relatively small number of graph nodes (≈100–150) is sufficient to capture the deformation patterns of the objects we consider, while also enabling very fast inference. We also experimented with denser graphs with more GNN nodes, but observed that inference becomes slower and the accuracy gain is marginal, so we adopt this node budget as a practical accuracy–efficiency trade-off.
> > >
> > > For more complex scenes with higher spatial resolution, GNN-based simulators have already been shown to scale to much larger graphs. For example, MeshGraphNets[4] trains on meshes with ~2k nodes and can be directly applied to meshes with ~20k nodes at test time. In addition, recent work on graph-based neural dynamics for contact-rich manipulation (e.g., CompNeRFdyn[5]) demonstrates that GNN-style models can effectively capture complex multi-object interactions and collisions in more challenging scenes. This suggests that, in principle, our GNN-based world model can also be extended to larger, multi-object, and more complex scenes, and we view a systematic study of this scaling behavior—including multi-object interactions and contact-rich scenarios—as important future work.
> > >
> > > Regarding explicit physical-law consistency metrics (e.g., momentum/energy conservation), our learned GNN operates on normalized features rather than explicit physical quantities (mass, energy), so directly measuring conservation inside the network is not straightforward. Instead, we provide evidence through real-motion fidelity and physically grounded supervision:
> > >
> > > * As reported in Table 1, our GNN predictions achieve lower Chamfer and Track losses against real-world object motions than the prior SOTA PhysTwin. Since real motions are naturally physically valid, matching them closely indicates strong physical realism.
> > >
> > > * Our MPM baseline also attains excellent prediction metrics, and MPM simulations explicitly enforce core laws such as mass and momentum conservation. Because our GNN-based world model is trained on demonstrations generated by this MPM simulator, the supervision signals are physically consistent, encouraging the learned dynamics to inherit these principles implicitly.
> > >
> > > Designing explicit conservation-based diagnostics and metrics tailored to neural world models is an interesting and complementary direction, and we will highlight this as future work.
> > >
> > > [4] Learning mesh-based simulation with graph networks. ICLR 2020.
> > >
> > > [5] Learning multi-object dynamics with compositional neural radiance field. CoRL 2022.
> > >
> > > ## Question 7: Training Sensitivity to Hyperparameters
> > >
> > > Response:
> > >
> > > To assess the robustness of our training procedure, we ran additional experiments with three different random seeds (leading to different parameter initializations) and varied the batch size and learning rate. We found that changing random seeds or batch size has negligible impact on final performance, and the training loss curves consistently converge to similar values. The learning rate is also stable within a reasonable range; however, when it is set too small, convergence becomes noticeably slower (though it still reaches a comparable minimum). This indicates that PhysWorld is not overly sensitive to initialization or most hyperparameters, with learning rate mainly affecting convergence speed. We have incorporated the above results into our revised manuscript in **Appendix D.4** (Figure 8, Figure 9, Figure 10).
> > >
> > > ## Question 8: Scope of the Term “World Model”
> > >
> > > Response:
> > >
> > > We appreciate the reviewer’s comment and agree that clarifying scope is important. In model-based RL/robotics, a “world model” typically refers to a learned forward dynamics model that predicts future states/observations from the current state and action, without necessarily covering full scene-level semantics. PhysWorld is such a model at the object level: given a deformable object’s current 3D state and an applied action, it predicts the object’s future 3D dynamics (and rendered observations), rather than all scene entities. In this common usage, an object-level dynamics model can also be called an object-level world model—a scope we already emphasize in the paper title—and we will explicitly clarify this terminology and limitation in the revised version of our paper.

---

> > > > ### Author Response · Authors · 2025-11-21
> > > >
> > > > ## Question 9: Transferability of Calibrated Physical Parameters
> > > >
> > > > Response:
> > > >
> > > > In our method, the calibrated physical parameters Φ are meant to be object-specific. This is because different objects naturally have different material and contact properties (e.g., stiffness, friction), so their true physical parameters should not be the same. If we directly reuse Φ from one object on another, it can mismatch the new object’s physics and lead to inaccurate predictions. For objects that are very similar (e.g., ropes with different thickness), we can warm-start Φ from a previously calibrated object and then fine-tune it using the new object’s videos. This can save time, but we do observe a small drop in accuracy compared to calibrating Φ from scratch for the new object. Exploring more systematic transfer of Φ or meta-learning good initializations across a family of objects is an interesting direction for future work.
> > > >
> > > > ## Question 10: Prospects for Explicit Physical Consistency Losses
> > > >
> > > > Response:
> > > >
> > > > We agree that explicit physical consistency losses could potentially further improve realism. In our current formulation, however, the GNN operates on learned, normalized feature representations rather than explicit physical quantities such as mass, momentum, or energy, which makes it non-trivial to design directly interpretable conservation losses. Instead, we rely on training data generated by an MPM simulator that already enforces fundamental physical principles (e.g., mass and momentum conservation), so the learned dynamics model is encouraged to respect these laws implicitly through supervision. Designing suitable proxies for quantities such as momentum or volume and incorporating them as additional regularizers in the loss function is an interesting and complementary direction, which we will highlight as future work.

---

### Author Response · Authors · 2025-12-02

**Dear Program Chairs, Senior Area Chairs, Area Chairs, and Reviewers**,

We sincerely thank you for your constructive feedback and expert insights. We took every concern seriously and the revision has included substantial updates. As the discussion phase nears its end, we would like to provide a concise summary of the key revisions and new experiments we have added during the rebuttal period to address the core concerns.

1. **Quantitative Evaluation on Downstream Planning Tasks (Reviewers YYty, WAMx)**

    To address the concern regarding the lack of quantitative control results, we have included quantitative evaluation of model-based planning using MPPI as shown in Figure 5, 6 and Table 9 in **Appendix D.1**. We reported distance error curves and success rates under varying thresholds and measured the wall-clock time for planning steps: **PhysWorld (22 s)** vs. PhysTwin (998 s) vs. MPM (4,687 s). The results demonstrate that our method requires significantly less planning time while maintaining satisfactory control performance.

2. **Clarification on Novelty and Contributions (Reviewer WAMx)**

    We further clarified the core innovative aspects of our framework-"**physically consistent yet diverse synthesis**". We introduce a unique perspective of ensuring physical consistency (via **VLM-assisted constitutive model selection** and **global-to-local physical property optimization**) while simultaneously maximizing diversity (via **various motion pattern generation (VMP-Gen)** and **part-aware physical property perturbation ($P^3$-Pert)**). This specific combination is critical for learning efficient world models from short videos (1-10s) and has not been explored in previous works.

3. **Architecture Comparison: Why GNN? (Reviewers YYty, aLrE)**

    In response to questions about why we chose GNNs over Transformers or MLPs, we implemented and trained variants based on PointMLP[1] and PointTransformerV3[2] as baselines. Table 7 in **Appendix C.1** now presents a comparative study. The results demonstrate that GNNs exhibit a stronger inductive bias for modeling sparse, local particle interactions in deformable objects, offering superior performance in both accuracy and efficiency.

    [1] Rethinking network design and local geometry in point cloud: A simple residual MLP framework. ICLR 2022.

    [2] Point transformer v3: Simpler faster stronger. CVPR 2024.

4. **Physical Realism Evaluation of Synthetic Demonstrations (Reviewers YYty, aLrE)**

    To address concerns about the physical plausibility of our synthetic demonstrations, we evaluated them using the PhyGenBench[3] protocol (checking 27 physical laws). Our data achieved a Realism Score of **1.6010**, highly consistent with real videos (**1.6087**).

    [3] Towards world simulator: Crafting physical commonsense-based benchmark for video generation. ICML 2025.

5. **Robustness of VLM-Assisted Constitutive Model Selection (Reviewers YYty, kicv, aLrE)**

    We also addressed concerns regarding the robustness of VLM-assisted constitutive model selection. We have provided a sensitivity analysis of the VLM-based constitutive model selection in Table 10 (in **Appendix D.3**). We demonstrate that the pipeline can predict accurate dynamics even if a suboptimal constitutive model is selected. We have also verified the VLM's selection accuracy (100%) on an extended dataset of internet videos shown in Figure 7.

6. **PhysTwin + GNN Distillation Experiments (Reviewer oNhM)**

    To address the suggestion of using PhysTwin as a teacher for distillation, we conducted specific comparative experiments. **Appendix C.2** has included a new ablation comparing "PhysTwin-distilled GNN" vs. "PhysWorld (MPM-distilled GNN)". The quantitative results are shown in Table 8. The results prove that our proposed **VMP-Gen** and **$P^3$-Pert** modules are crucial for generalization even when using PhysTwin as the teacher model, and that our MPM-based approach serves as a superior teacher compared to PhysTwin.

7. **Additional Evaluation on Elasto-plastic Objects (Reviewers oNhM, WAMx)**

    Addressing the concern that the tested scenes were limited to simple deformables, Table 12 and Figure 11 in **Appendix D.6** demonstrate that our calibrated MPM digital twin significantly outperforms PhysTwin (spring-mass) on elasto-plastic objects from RoboCraft[4] involving irreversible deformations. This validates the methodological necessity of using MPM over spring-mass systems for capturing complex, non-reversible physical behaviors.

    [4] Robocraft: Learning to see, simulate, and shape elasto-plastic objects in 3d with graph networks. RSS 2022.

We have incorporated all the above additions into the revised manuscript. We believe these supplementary experiments and analyses effectively address the reviewers' concerns and solidify the paper's contribution.

Thank you again for your time and dedication to improving our work.

Sincerely,

**The Authors of Submission #6146**

---

### Meta-Review · Area_Chair_GfcQ · 2026-01-14

**Summary:**

The reviewers’ concerns mainly focus on clarifying the novelty and scientific insights of the proposed method, conducting more thorough architectural ablations and downstream benchmarking, analyzing the overall training complexity, evaluating the robustness of the approach, and providing quantitative assessments on real-world synthesis tasks. The rebuttal provides additional experimental results and quantitative evaluations to address these concerns. However, the reported computational cost indicates that the training process remains non-trivial. Moreover, some reviewers continue to raise concerns regarding the paper’s validation. The paper could be further strengthened by considering tasks with clearer contextual relevance and more substantial challenges beyond reconfiguring simple deformable structures.

**Reviewer Concerns:**

The authors provide sufficient rebuttal material to address most of the reviewers’ concerns. Based on the presented results, the concerns raised by reviewers YYty, aLrE, and oNhM appear to be well addressed. However, during the rebuttal period, the official comments from reviewer WAMx show that his/her concerns are not fully answered. For reviewer kicv, it remains unclear whether the provided summary of computational cost sufficiently meets the stated requirements.

**Reviewer Scores:**

1. YYty: The reviewer may raise the score, as the additional downstream benchmark experiments and quantitative evaluations further substantiate the effectiveness of the proposed method. Moreover, the point-by-point rebuttal adequately addresses all raised concerns.
2. kicv: The reviewer is likely to maintain the current score, since the reported computational cost indicates that the training time remains non-trivial.
3. WAMx: The reviewer is expected to maintain the score. In the rebuttal, the reviewer explicitly notes that “the paper's validation, including the appended one, looks slim,” suggesting that the concerns were not fully resolved.
4. aLrE: The reviewer may raise the score, as the additional results provide further positive evidence in support of the paper.
5. oNhM: The reviewer may raise the score, as the additional results provide positive evidence supporting the paper.

---

### Decision · Program_Chairs · 2026-01-26

Reject